# Dream the Impossible:
# Outlier Imagination with Diffusion Models

**Xuefeng Du, Yiyou Sun, Xiaojin Zhu, Yixuan Li**
Department of Computer Sciences
University of Wisconsin, Madison
{xfdu,sunyiyou,jerryzhu,sharonli}@cs.wisc.edu

## Abstract

Utilizing auxiliary outlier datasets to regularize the machine learning model has demonstrated promise for out-of-distribution (OOD) detection and safe prediction. Due to the labor intensity in data collection and cleaning, automating outlier data generation has been a long-desired alternative. Despite the appeal, generating photo-realistic outliers in the high dimensional pixel space has been an open challenge for the field. To tackle the problem, this paper proposes a new framework DREAM-OOD, which enables imagining photo-realistic outliers by way of diffusion models, provided with only the in-distribution (ID) data and classes. Specifically, DREAM-OOD learns a text-conditioned latent space based on ID data, and then samples outliers in the low-likelihood region via the latent, which can be decoded into images by the diffusion model. Different from prior works [1, 2], DREAM-OOD enables visualizing and understanding the imagined outliers, directly in the pixel space. We conduct comprehensive quantitative and qualitative studies to understand the efficacy of DREAM-OOD, and show that training with the samples generated by DREAM-OOD can benefit OOD detection performance. Code is publicly available at https://github.com/deeplearning-wisc/dream-ood.

## 1 Introduction

Out-of-distribution (OOD) detection is critical for deploying machine learning models in the wild, where samples from novel classes can naturally emerge and should be flagged for caution. Concerningly, modern neural networks are shown to produce overconfident and therefore untrustworthy predictions for unknown OOD inputs [3]. To mitigate the issue, recent works have explored training with an auxiliary outlier dataset, where the model is regularized to learn a more conservative decision boundary around in-distribution (ID) data [4–7]. These methods have demonstrated encouraging OOD detection performance over the counterparts without auxiliary data.

Despite the promise, preparing auxiliary data can be labor-intensive and inflexible, and necessitates careful human intervention, such as data cleaning, to ensure the auxiliary outlier data does not overlap with the ID data. Automating outlier data generation has thus been a long-desired alternative. Despite the appeal, generating photo-realistic outliers has been extremely challenging due to the high dimensional space. Recent works including VOS and NPOS [1, 2] proposed sampling outliers in the low-dimensional feature space and directly employed the latent-space outliers to regularize the model. However, these latent-space methods do not allow us to understand the outliers in a human-compatible way. Today, the field still lacks an automatic mechanism to generate high-resolution outliers in the *pixel space*.

In this paper, we propose a new framework DREAM-OOD that enables imagining photo-realistic outliers by way of diffusion models, provided with only ID data and classes (see Figure 1). Harnessing the power of diffusion models for outlier imagination is non-trivial, since one cannot easily describe

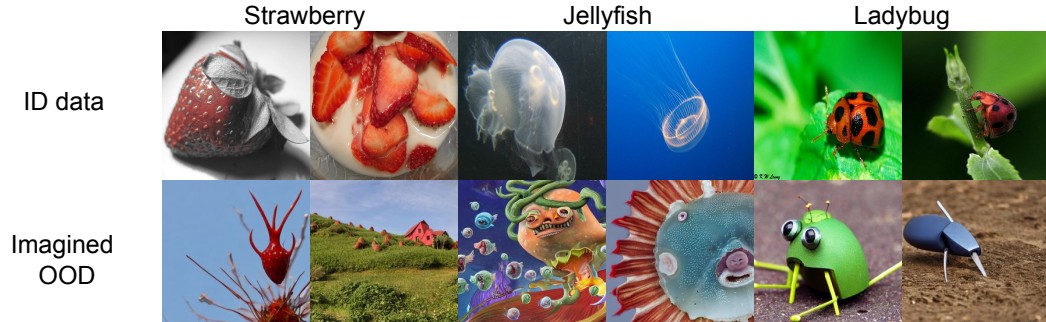

Figure 1: **Top**: Original ID training data in IMAGENET [8]. **Bottom**: Samples generated by our method DREAM-OOD, which deviate from the ID data.

the exponentially many possibilities of outliers using text prompts. It can be particularly challenging to characterize informative outliers that lie on the boundary of ID data, which have been shown to be the most effective in regularizing the ID classifier and its decision boundary [7]. After all, it is almost impossible to describe something in words without knowing what it looks like.

Our framework circumvents the above challenges by: **(1)** learning compact visual representations for the ID data, conditioned on the textual latent space of the diffusion model (Section 3.1), and **(2)** sampling new visual embeddings in the text-conditioned latent space, which are then decoded to pixel-space images by the diffusion model (Section 3.2). Concretely, to learn the text-conditioned latent space, we train an image classifier to produce image embeddings that have a higher probability to be aligned with the corresponding class token embedding. The resulting feature embeddings thus form a compact and informative distribution that encodes the ID data. Equipped with the text-conditioned latent space, we sample new embeddings from the low-likelihood region, which can be decoded into the images via the diffusion model. The rationale is if the sampled embedding is distributionally far away from the in-distribution embeddings, the generated image will have a large semantic discrepancy from the ID images and vice versa.

We demonstrate that our proposed framework creatively imagines OOD samples conditioned on a given dataset, and as a result, helps improve the OOD detection performance. On IMAGENET dataset, training with samples generated by DREAM-OOD improves the OOD detection on a comprehensive suite of OOD datasets. Different from [1, 2], our method allows visualizing and understanding the imagined outliers, covering a wide spectrum of near-OOD and far-OOD. Note that DREAM-OOD enables leveraging off-the-shelf diffusion models for OOD detection, rather than modifying the diffusion model (which is an actively studied area on its own [9]). In other words, this work's core contribution is to leverage generative modeling to improve discriminative learning, establishing innovative connections between the diffusion model and outlier data generation.

Our key contributions are summarized as follows:

1. To the best of our knowledge, DREAM-OOD is the first to enable the generation of photo-realistic high-resolution outliers for OOD detection. DREAM-OOD establishes promising performance on common benchmarks and can benefit OOD detection.

2. We conduct comprehensive analyses to understand the efficacy of DREAM-OOD, both quantitatively and qualitatively. The results provide insights into outlier imagination with diffusion models.

3. As an *extension*, we show that our synthesis method can be used to automatically generate ID samples, and as a result, improves the generalization performance of the ID task itself.

## 2   Preliminaries

We consider a training set $\mathcal{D} = \{(\mathbf{x}_i, y_i)\}_{i=1}^n$, drawn *i.i.d.* from the joint data distribution $P_{\mathcal{X}\mathcal{Y}}$. $\mathcal{X}$ denotes the input space and $\mathcal{Y} \in \{1, 2, ..., C\}$ denotes the label space. Let $\mathbb{P}_{\text{in}}$ denote the marginal distribution on $\mathcal{X}$, which is also referred to as the *in-distribution*. Let $f_\theta : \mathcal{X} \mapsto \mathbb{R}^C$ denote a multi-class classifier, which predicts the label of an input sample with parameter $\theta$. To obtain an

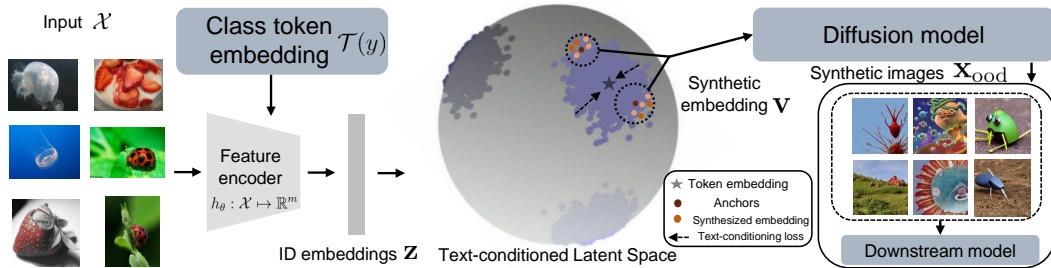

Figure 2: **Illustration of our proposed outlier imagination framework DREAM-OOD.** DREAM-OOD first learns a text-conditioned space to produce compact image embeddings aligned with the token embedding $\mathcal{T}(y)$ of the diffusion model. It then samples new embeddings in the latent space, which can be decoded into pixel-space outlier images $\mathbf{x}_{\text{ood}}$ by diffusion model. The newly generated samples can help improve OOD detection. Best viewed in color.

optimal classifier $f^*$, a standard approach is to perform empirical risk minimization (ERM) [10]: $f^* = \text{argmin}_{f \in \mathcal{F}} \frac{1}{n} \sum_{i=1}^{n} \ell(f(\mathbf{x}_i), y_i)$ where $\ell$ is the loss function and $\mathcal{F}$ is the hypothesis space.

**Out-of-distribution detection.** When deploying a machine model in the real world, a reliable classifier should not only accurately classify known in-distribution samples, but also identify OOD input from *unknown* class $y \notin \mathcal{Y}$. This can be achieved by having an OOD detector, in tandem with the classification model $f_\theta$. At its core, OOD detection can be formulated as a binary classification problem. At test time, the goal is to decide whether a test-time input is from ID or not (OOD). We denote $g_\theta : \mathcal{X} \mapsto \{\text{in}, \text{out}\}$ as the function mapping for OOD detection.

**Denoising diffusion models** have emerged as a promising generative modeling framework, pushing the state-of-the-art in image generation [11, 12]. Inspired by non-equilibrium thermodynamics, diffusion probabilistic models [13–15] define a forward Gaussian Markov transition kernel of diffusion steps to gradually corrupt training data until the data distribution is transformed into a simple noisy distribution. The model then learns to reverse this process by learning a denoising transition kernel parameterized by a neural network.

Diffusion models can be conditional, for example, on class labels or text descriptions [11, 16, 17]. In particular, Stable Diffusion [18] is a text-to-image model that enables synthesizing new images guided by the text prompt. The model was trained on 5 billion pairs of images and captions taken from LAION-5B [19], a publicly available dataset derived from Common Crawl data scraped from the web. Given a class name $y$, the generation process can be mathematically denoted by:

$$\mathbf{x} \sim P(\mathbf{x}|\mathbf{z}_y), \tag{1}$$

where $\mathbf{z}_y = \mathcal{T}(y)$ is the textual representation of label $y$ with prompting (e.g., "A high-quality photo of a [y]"). In Stable Diffusion, $\mathcal{T}(\cdot)$ is the text encoder of the CLIP model [20].

## 3   DREAM-OOD: Outlier Imagination with Diffusion Models

In this paper, we propose a novel framework that enables synthesizing photo-realistic outliers with respect to a given ID dataset (see Figure 1). The synthesized outliers can be useful for regularizing the ID classifier to be less confident in the OOD region. Recall that the vanilla diffusion generation takes as input the textual representation. While it is easy to encode the ID classes $y \in \mathcal{Y}$ into textual latent space via $\mathcal{T}(y)$, one cannot trivially generate text prompts for outliers. It can be particularly challenging to characterize informative outliers that lie on the boundary of ID data, which have been shown to be most effective in regularizing the ID classifier and its decision boundary [7]. After all, it is almost impossible to concretely describe something in words without knowing what it looks like.

**Overview.** As illustrated in Figure 2, our framework circumvents the challenge by: **(1)** learning compact visual representations for the ID data, conditioned on the textual latent space of the diffusion model (Section 3.1), and **(2)** sampling new visual embeddings in the text-conditioned latent space, which are then decoded into the images by diffusion model (Section 3.2). We demonstrate in Section 4 that, our proposed outlier synthesis framework produces meaningful out-of-distribution samples conditioned on a given dataset, and as a result, significantly improves the OOD detection performance.

## 3.1 Learning the Text-Conditioned Latent Space

Our key idea is to first train a classifier on ID data $\mathcal{D}$ that produces image embeddings, conditioned on the token embeddings $\mathcal{T}(y)$, with $y \in \mathcal{Y}$. To learn the text-conditioned visual latent space, we train the image classifier to produce image embeddings that have a higher probability of being aligned with the corresponding class token embedding, and vice versa.

Specifically, denote $h_\theta : \mathcal{X} \mapsto \mathbb{R}^m$ as a feature encoder that maps an input $\mathbf{x} \in \mathcal{X}$ to the image embedding $h_\theta(\mathbf{x})$, and $\mathcal{T} : \mathcal{Y} \mapsto \mathbb{R}^m$ as the text encoder that takes a class name $y$ and outputs its token embedding $\mathcal{T}(y)$. Here $\mathcal{T}(\cdot)$ is a fixed text encoder of the diffusion model. Only the image feature encoder needs to be trained, with learnable parameters $\theta$. Mathematically, the loss function for learning the visual representations is formulated as follows:

$$\mathcal{L} = \mathbb{E}_{(\mathbf{x},y)\sim\mathcal{D}}[-\log \frac{\exp\left(\mathcal{T}(y)^\top \mathbf{z}/t\right)}{\sum_{j=1}^{C} \exp\left(\mathcal{T}(y_j)^\top \mathbf{z}/t\right)}], \quad (2)$$

where $\mathbf{z} = h_\theta(\mathbf{x})/\|h_\theta(\mathbf{x})\|_2$ is the $L_2$-normalized image embedding, and $t$ is temperature.

**Theoretical interpretation of loss.** Formally, our loss function directly promotes the class-conditional von Mises Fisher (vMF) distribution [21–23]. vMF is analogous to spherical Gaussian distributions for features with unit norms ($\|\mathbf{z}\|^2 = 1$). The probability density function of $\mathbf{z} \in \mathbb{R}^m$ in class $c$ is:

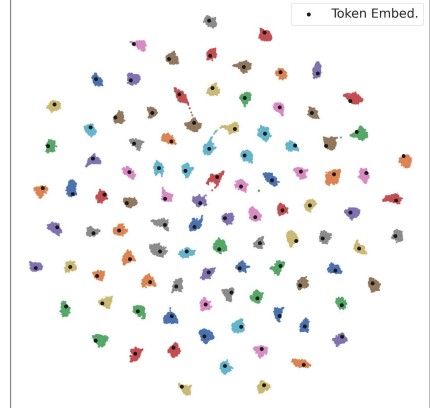

Figure 3: **TSNE visualization of learned feature embeddings using $\mathcal{L}$.** Black dots indicate token embeddings, one for each class.

$$p_m(\mathbf{z}; \boldsymbol{\mu}_c, \kappa) = Z_m(\kappa) \exp\left(\kappa \boldsymbol{\mu}_c^\top \mathbf{z}\right), \quad (3)$$

where $\boldsymbol{\mu}_c$ is the class centroid with unit norm, $\kappa \geq 0$ controls the extent of class concentration, and $Z_m(\kappa)$ is the normalization factor detailed in the Appendix B. The probability of the feature vector $\mathbf{z}$ belonging to class $c$ is:

$$
\begin{aligned}
P(y = c | \mathbf{z}; \{\kappa, \boldsymbol{\mu}_j\}_{j=1}^C) &= \frac{Z_m(\kappa) \exp\left(\kappa \boldsymbol{\mu}_c^\top \mathbf{z}\right)}{\sum_{j=1}^{C} Z_m(\kappa) \exp\left(\kappa \boldsymbol{\mu}_j^\top \mathbf{z}\right)} \\
&= \frac{\exp\left(\boldsymbol{\mu}_c^\top \mathbf{z}/t\right)}{\sum_{j=1}^{C} \exp\left(\boldsymbol{\mu}_j^\top \mathbf{z}/t\right)},
\end{aligned}
\quad (4)
$$

where $\kappa = \frac{1}{t}$. Therefore, by encouraging features to be aligned with its class token embedding, our loss function $\mathcal{L}$ (Equation (2)) maximizes the log-likelihood of the class-conditional vMF distributions and promotes compact clusters on the hypersphere (see Figure 3). The highly compact representations can benefit the sampling of new embeddings, as we introduce next in Section 3.2.

## 3.2 Outlier Imagination via Text-Conditioned Latent

Given the well-trained compact representation space that encodes the information of $\mathbb{P}_{\text{in}}$, we propose to generate outliers by sampling new embeddings in the text-conditioned latent space, and then decoding via diffusion model. The rationale is that if the sampled embeddings are distributionally far away from the ID embeddings, the decoded images will have a large semantic discrepancy with the ID images and vice versa.

Recent works [1, 2] proposed sampling outlier embeddings and directly employed the latent-space outliers to regularize the model. In contrast, our method focuses on generating *pixel-space* photo-realistic images, which allows us to directly inspect the generated outliers in a human-compatible way. Despite the appeal, generating high-resolution outliers has been extremely challenging due to the high dimensional space. To tackle the issue, our generation procedure constitutes two steps:

1. *Sample OOD in the latent space*: draw new embeddings $\mathbf{v}$ that are in the low-likelihood region of the text-conditioned latent space.

2. *Image generation:* decode $\mathbf{v}$ into a pixel-space OOD image via diffusion model.

**Algorithm 1** DREAM-OOD: Outlier Imagination with Diffusion Models

---

**Input:** In-distribution training data $\mathcal{D} = \{(\mathbf{x}_i, y_i)\}_{i=1}^n$, initial model parameters $\theta$ for learning the text-conditioned latent space, diffusion model.
**Output:** Synthetic images $\mathbf{x}_{\text{ood}}$.
**Phases:** Phase 1: Learning the Text-conditioned Latent Space. Phase 2: Outlier Imagination via Text-Conditioned Latent.
**while** *Phase 1* **do**
 |  1. Extract token embeddings $\mathcal{T}(y)$ of the ID label $y \in \mathcal{Y}$.
 |  2. Learn the text-conditioned latent representation space by Equation (2).
**end**
**while** *Phase 2* **do**
 |  1. Sample a set of outlier embeddings $V_i$ in the low-likelihood region of the text-conditioned latent space as in Section 3.2.
 |  2. Decode the outlier embeddings into the pixel-space OOD images via diffusion model by Equation (6).
**end**

---

**Sampling OOD embedding.** Our goal here is to sample low-likelihood embeddings based on the learned feature representations (see Figure 4). The sampling procedure can be instantiated by different approaches. For example, a recent work by Tao *et.al.* [2] proposed a latent non-parametric sampling method, which does not make any distributional assumption on the ID embeddings and offers stronger flexibility compared to the parametric sampling approach [1]. Concretely, we can select the boundary ID anchors by leveraging the non-parametric nearest neighbor distance, and then draw new embeddings around that boundary point.

Denote the $L_2$-normalized embedding set of training data as $\mathbb{Z} = (\mathbf{z}_1, \mathbf{z}_2, ..., \mathbf{z}_n)$, where $\mathbf{z}_i = h_\theta(\mathbf{x}_i)/\|h_\theta(\mathbf{x}_i)\|_2$. For any embedding $\mathbf{z}' \in \mathbb{Z}$, we calculate the $k$-NN distance *w.r.t.* $\mathbb{Z}$:

$$d_k(\mathbf{z}', \mathbb{Z}) = \|\mathbf{z}' - \mathbf{z}_{(k)}\|_2, \tag{5}$$

where $\mathbf{z}_{(k)}$ is the $k$-th nearest neighbor in $\mathbb{Z}$. If an embedding has a large $k$-NN distance, it is likely to be on the boundary of the ID data and vice versa.

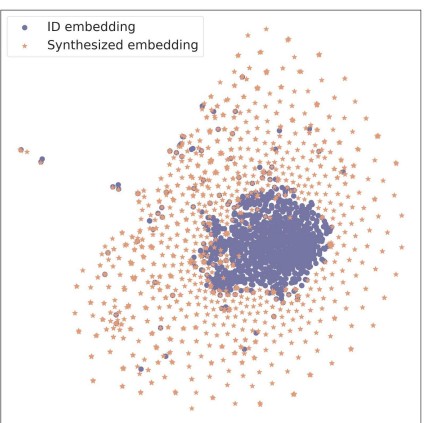

Given a boundary ID point, we then draw new embedding sample $\mathbf{v} \in \mathbb{R}^m$ from a Gaussian kernel[1] centered at $\mathbf{z}_i$ with covariance $\sigma^2 \mathbf{I}$: $\mathbf{v} \sim \mathcal{N}(\mathbf{z}_i, \sigma^2 \mathbf{I})$. In addition, to ensure that the outliers are sufficiently far away from the ID data, we repeatedly sample multiple outlier embeddings from the Gaussian kernel $\mathcal{N}(\mathbf{z}_i, \sigma^2 \mathbf{I})$, which produces a set $V_i$, and further perform a filtering process by selecting the outlier embedding in $V_i$ with the largest $k$-NN distance *w.r.t.* $\mathbb{Z}$. Detailed ablations on the sampling parameters are provided in Section 4.2.

Figure 4: TSNE visualization of ID embeddings (purple) and the sampled outlier embeddings (orange), for the class "hen" in IMAGENET.

**Outlier image generation.** Lastly, to obtain the outlier images in the pixel space, we decode the sampled outlier embeddings $\mathbf{v}$ via the diffusion model. In practice, this can be done by replacing the original token embedding $\mathcal{T}(y)$ with the sampled new embedding $\mathbf{v}$[2]. Different from the vanilla prompt-based generation (*c.f.* Equation (1)) , our outlier imagination is mathematically reflected by:

$$\mathbf{x}_{\text{ood}} \sim P(\mathbf{x}|\mathbf{v}), \tag{6}$$

where $\mathbf{x}_{\text{ood}}$ denotes the generated outliers in the pixel space. Importantly, $\mathbf{v} \sim S \circ h_\theta \circ (\mathbb{P}_{\text{in}})$ is dependent on the in-distribution data, which enables generating images that deviate from $\mathbb{P}_{\text{in}}$. $S(\cdot)$ denotes the sampling procedure. Our framework DREAM-OOD is summarized in Algorithm 1.

---

[1]The choice of kernel function form (*e.g.*, Gaussian vs. Epanechnikov) is not influential, while the kernel bandwidth parameter is [24].

[2]In the implementation, we re-scale $\mathbf{v}$ by multiplying the norm of the original token embedding to preserve the magnitude.

| Methods | OOD Datasets | | | | | | | | | | ID ACC |
| | iNaturalist | | Places | | Sun | | Textures | | Average | | |
| | FPR95↓ | AUROC↑ | FPR95↓ | AUROC↑ | FPR95↓ | AUROC↑ | FPR95↓ | AUROC↑ | FPR95↓ | AUROC↑ | |
| MSP [25] | 31.80 | 94.98 | 47.10 | 90.84 | 47.60 | 90.86 | 65.80 | 83.34 | 48.08 | 90.01 | 87.64 |
| ODIN [26] | 24.40 | 95.92 | 50.30 | 90.20 | 44.90 | 91.55 | 61.00 | 81.37 | 45.15 | 89.76 | 87.64 |
| Mahalanobis [27] | 91.60 | 75.16 | 96.70 | 60.87 | 97.40 | 62.23 | 36.50 | 91.43 | 80.55 | 72.42 | 87.64 |
| Energy [6] | 32.50 | 94.82 | 50.80 | 90.76 | 47.60 | 91.71 | 63.80 | 80.54 | 48.68 | 89.46 | 87.64 |
| GODIN [28] | 39.90 | 93.94 | 59.70 | 89.20 | 58.70 | 90.65 | 39.90 | 92.71 | 49.55 | 91.62 | 87.38 |
| KNN [29] | 28.67 | 95.57 | 65.83 | 88.72 | 58.08 | 90.17 | 12.92 | 90.37 | 41.38 | 91.20 | 87.64 |
| ViM [30] | 75.50 | 87.18 | 88.30 | 81.25 | 88.70 | 81.37 | 15.60 | 96.63 | 67.03 | 86.61 | 87.64 |
| ReAct [31] | 22.40 | 96.05 | 45.10 | 92.28 | 37.90 | 93.04 | 59.30 | 85.19 | 41.17 | 91.64 | 87.64 |
| DICE [32] | 37.30 | 92.51 | 53.80 | 87.75 | 45.60 | 89.21 | 50.00 | 83.27 | 46.67 | 88.19 | 87.64 |
| *Synthesis-based methods* | | | | | | | | | | | |
| GAN [33] | 83.10 | 71.35 | 83.20 | 69.85 | 84.40 | 67.56 | 91.00 | 59.16 | 85.42 | 66.98 | 79.52 |
| VOS [1] | 43.00 | 93.77 | 47.60 | 91.77 | 39.40 | 93.17 | 66.10 | 81.42 | 49.02 | 90.03 | 87.50 |
| NPOS [2] | 53.84 | 86.52 | 59.66 | 83.50 | 53.54 | 87.99 | **8.98** | **98.13** | 44.00 | 89.04 | 85.37 |
| **DREAM-OOD (Ours)** | **24.10**±0.2 | **96.10**±0.1 | **39.87**±0.1 | **93.11**±0.3 | **36.88**±0.4 | **93.31**±0.4 | 53.99±0.6 | 85.56±0.9 | **38.76**±0.2 | **92.02**±0.4 | 87.54±0.1 |

Table 1: OOD detection results for IMAGENET-100 as the in-distribution data. We report standard deviations estimated across 3 runs. Bold numbers are superior results.

**Learning with imagined outlier images.** The generated synthetic OOD images $\mathbf{x}_{ood}$ can be used for regularizing the training of the classification model [1]:

$$\mathcal{L}_{ood} = \mathbb{E}_{\mathbf{x}_{ood}}\left[-\log\frac{1}{1+\exp^{\phi(E(f_\theta(\mathbf{x}_{ood})))}}\right] + \mathbb{E}_{\mathbf{x}\sim\mathbb{P}_{in}}\left[-\log\frac{\exp^{\phi(E(f_\theta(\mathbf{x})))}}{1+\exp^{\phi(E(f_\theta(\mathbf{x})))}}\right], \qquad (7)$$

where $\phi(\cdot)$ is a three-layer nonlinear MLP function with the same architecture as VOS [1], $E(\cdot)$ denotes the energy function, and $f_\theta(\mathbf{x})$ denotes the logit output of the classification model. In other words, the loss function takes both the ID and generated OOD images, and learns to separate them explicitly. The overall training objective combines the standard cross-entropy loss, along with an additional loss in terms of OOD regularization $\mathcal{L}_{CE} + \beta \cdot \mathcal{L}_{ood}$, where $\beta$ is the weight of the OOD regularization. $\mathcal{L}_{CE}$ denotes the cross-entropy loss on the ID training data. In testing, we use the output of the binary logistic classifier for OOD detection.

## 4 Experiments and Analysis

In this section, we present empirical evidence to validate the effectiveness of our proposed outlier imagination framework. In what follows, we show that DREAM-OOD produces meaningful OOD images, and as a result, significantly improves OOD detection (Section 4.1) performance. We provide comprehensive ablations and qualitative studies in Section 4.2. In addition, we showcase an *extension* of our framework for improving generalization by leveraging the synthesized inliers (Section 4.3).

### 4.1 Evaluation on OOD Detection Performance

**Datasets.** Following [2], we use the CIFAR-100 and the large-scale IMAGENET dataset [8] as the ID training data. For CIFAR-100, we use a suite of natural image datasets as OOD including TEXTURES [34], SVHN [35], PLACES365 [36], iSUN [37] & LSUN [38]. For IMAGENET-100, we adopt the OOD test data as in [39], including subsets of iNATURALIST [40], SUN [41], PLACES [36], and TEXTURES [34]. For each OOD dataset, the categories are disjoint from the ID dataset. We provide the details of the datasets and categories in Appendix A.

**Training details.** We use ResNet-34 [42] as the network architecture for both CIFAR-100 and IMAGENET-100 datasets. We train the model using stochastic gradient descent for 100 epochs with the cosine learning rate decay schedule, a momentum of 0.9, and a weight decay of $5e^{-4}$. The initial learning rate is set to 0.1 and the batch size is set to 160. We generate $1,000$ OOD samples per class using Stable Diffusion v1.4, which results in $100,000$ synthetic images in total. $\beta$ is set to 1.0 for IMAGENET-100 and 2.5 for CIFAR-100. To learn the feature encoder $h_\theta$, we set the temperature $t$ in Equation (2) to 0.1. Extensive ablations on hyperparameters $\sigma$, $k$ and $\beta$ are provided in Section 4.2.

**Evaluation metrics.** We report the following metrics: (1) the false positive rate (FPR95) of OOD samples when the true positive rate of ID samples is 95%, (2) the area under the receiver operating characteristic curve (AUROC), and (3) ID accuracy (ID ACC).

**DREAM-OOD significantly improves the OOD detection performance.** As shown in Table 1 and Table 3, we compare our method with the competitive baselines, including Maximum Softmax Probability [25], ODIN score [26], Mahalanobis score [27], Energy score [6], Generalized ODIN [28],

| Methods | OOD Datasets | | | | | | | | | | | | ID ACC |
|---|---|---|---|---|---|---|---|---|---|---|---|---|---|
| | SVHN | | PLACES365 | | LSUN | | iSUN | | TEXTURES | | Average | | |
| | FPR95↓ | AUROC↑ | FPR95↓ | AUROC↑ | FPR95↓ | AUROC↑ | FPR95↓ | AUROC↑ | FPR95↓ | AUROC↑ | FPR95↓ | AUROC↑ | |
| MSP [25] | 87.35 | 69.08 | 81.65 | 76.71 | 76.40 | 80.12 | 76.00 | 78.90 | 79.35 | 77.43 | 80.15 | 76.45 | 79.04 |
| ODIN [26] | 90.95 | 64.36 | 79.30 | 74.87 | 75.60 | 78.04 | 53.10 | 87.40 | 72.60 | 79.82 | 74.31 | 76.90 | 79.04 |
| Mahalanobis [27] | 87.80 | 69.98 | 76.00 | 77.90 | 56.80 | 85.83 | 59.20 | 86.46 | 62.45 | 84.43 | 68.45 | 80.92 | 79.04 |
| Energy [6] | 84.90 | 70.90 | 82.05 | 76.00 | 81.75 | 78.36 | 73.55 | 81.20 | 78.70 | 78.87 | 80.19 | 77.07 | 79.04 |
| GODIN [28] | 63.95 | 88.98 | 80.65 | 77.19 | 60.65 | 88.36 | 51.60 | 92.07 | 71.75 | 85.02 | 65.72 | 86.32 | 76.34 |
| KNN [29] | 81.12 | 73.65 | 79.62 | 78.21 | 63.29 | 85.56 | 73.92 | 79.77 | 73.29 | 80.35 | 74.25 | 79.51 | 79.04 |
| ViM [30] | 81.20 | 77.24 | 79.20 | 77.81 | 43.10 | 90.43 | 74.55 | 83.02 | 61.85 | 85.57 | 67.98 | 82.81 | 79.04 |
| ReAct [31] | 82.85 | 70.12 | 81.75 | 76.25 | 80.70 | 83.03 | 67.40 | 83.28 | 74.60 | 81.61 | 77.46 | 78.86 | 79.04 |
| DICE [32] | 83.55 | 72.49 | 85.05 | 75.92 | 94.05 | 73.59 | 75.20 | 80.90 | 79.80 | 77.83 | 83.53 | 76.15 | 79.04 |
| *Synthesis-based methods* | | | | | | | | | | | | | |
| GAN [33] | 89.45 | 66.95 | 88.75 | 66.76 | 82.35 | 75.87 | 83.45 | 73.49 | 92.80 | 62.99 | 87.36 | 69.21 | 70.12 |
| VOS [1] | 78.50 | 73.11 | 84.55 | 75.85 | 59.05 | 85.72 | 72.45 | 82.66 | 75.35 | 80.08 | 73.98 | 79.48 | 78.56 |
| NPOS [2] | 11.14 | 97.84 | 79.08 | 71.30 | 56.27 | 82.43 | 51.72 | 85.48 | 35.20 | 92.44 | 46.68 | 85.90 | 78.23 |
| **DREAM-OOD (Ours)** | 58.75±0.6 | 87.01±0.1 | 70.85±1.6 | 79.94±0.2 | 24.25±1.1 | 95.23±0.2 | 1.10±0.2 | 99.73±0.4 | 46.60±0.4 | 88.82±0.7 | 40.31±0.8 | 90.15±0.3 | 78.94 |

Table 3: OOD detection results for CIFAR-100 as the in-distribution data. We report standard deviations estimated across 3 runs. Bold numbers are superior results.

KNN distance [29], ViM score [30], ReAct [31], and DICE [32]. Closely related to ours, we contrast with three synthesis-based methods, including latent-based outlier synthesis (VOS [1] & NPOS [2]), and GAN-based synthesis [33], showcasing the effectiveness of our approach. For example, DREAM-OOD achieves an FPR95 of 39.87% on PLACES with the ID data of IMAGENET-100, which is a 19.79% improvement from the best baseline NPOS.

In particular, DREAM-OOD advances both VOS and NPOS by allowing us to understand the synthesized outliers in a human-compatible way, which was infeasible for the feature-based outlier sampling in VOS and NPOS. Compared with the feature-based synthesis approaches, DREAM-OOD can generate high-resolution outliers in the pixel space. The higher-dimensional pixel space offers much more knowledge about the unknowns, which provides the model with high variability and fine-grained details for the unknowns that are missing in VOS and NPOS. Since DREAM-OOD is more photo-realistic and better for humans, the generated images can be naturally better constrained for neural networks (for example, things may be more on the natural image manifolds). We provide comprehensive qualitative results (Section 4.2) to facilitate the understanding of generated outliers. As we will show in Figure 5, the generated outliers are more precise in characterizing OOD data and thus improve the empirical performance.

**Comparison with other outlier synthesis approaches.** We compare DREAM-OOD with different outlier embedding synthesis approaches in Table 2: (I) synthesizing outlier embeddings by adding multivariate Gaussian noise $\mathcal{N}(\mathbf{0}, \sigma_1^2 \mathbf{I})$ to the token embeddings, (II) adding learnable noise to the token embeddings where the

| Method | FPR95 ↓ | AUROC ↑ | FPR95 ↓ | AUROC ↑ |
|---|---|---|---|---|
| | IMAGENET-100 as ID | | CIFAR-100 as ID | |
| (I) Add gaussian noise | 41.35 | 89.91 | 45.33 | 88.83 |
| (II) Add learnable noise | 42.48 | 91.45 | 48.05 | 87.72 |
| (III) Interpolate embeddings | 41.35 | 90.82 | 43.36 | 87.09 |
| (IV) Disjont class names | 43.55 | 87.84 | 49.89 | 85.87 |
| **DREAM-OOD (ours)** | **38.76** | **92.02** | **40.31** | **90.15** |

Table 2: Comparison of DREAM-OOD with different outlier embedding synthesis methods using diffusion models.

noise is trained to push the outliers away from ID features, (III) interpolating token embeddings from different classes by $\alpha \mathcal{T}(y_1) + (1-\alpha)\mathcal{T}(y_2)$, and (IV) generating outlier images by using embeddings of new class names outside ID classes. For (I), we set the optimal variance values $\sigma_1^2$ to 0.03 by sweeping from $\{0.01, 0.02, 0.03, ..., 0.10\}$. For (III), we choose the interpolation factor $\alpha$ to be 0.5 from $\{0.1, 0.2, ..., 0.9\}$. For (IV), we use the remaining 900 classes in IMAGENET-1K (exclude the 100 classes in IMAGENET-100) as the disjoint class names for outlier generation. We generate the same amount of images as ours for all the variants to ensure a fair comparison.

The result shows that DREAM-OOD outperforms all the alternative synthesis approaches by a considerable margin. Though adding noise to the token embedding is relatively simple, it cannot explicitly sample textual embeddings from the low-likelihood region as DREAM-OOD does, which are near the ID boundary and thus demonstrate stronger effectiveness to regularize the model (Section 3.2). Visualization is provided in Appendix E. Interpolating the token embeddings will easily generate images that are still ID (Appendix D), which is also observed in Liew *et.al.* [43].

## 4.2 Ablation Studies

In this section, we provide additional ablations to understand DREAM-OOD for OOD generation. For all the ablations, we use the high resolution IMAGENET-100 dataset as the ID data.

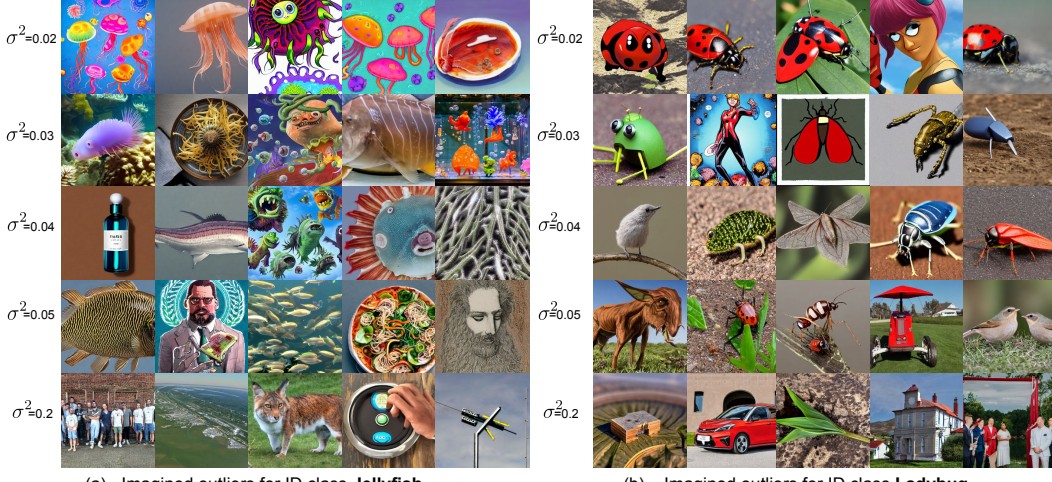

| (a) Imagined outliers for ID class **Jellyfish** | (b) Imagined outliers for ID class **Ladybug** |

Figure 5: **Visualization of the imagined outliers** *w.r.t. jellyfish, ladybug* class under different variance $\sigma^2$.

**Ablation on the regularization weight $\beta$.** In Figure 7 (a), we ablate the effect of weight $\beta$ of the regularization loss $\mathcal{L}_{\text{ood}}$ for OOD detection (Section 3.2) on the OOD detection performance. Using a mild weighting, such as $\beta = 1.0$, achieves the best OOD detection performance. Too excessive regularization using synthesized OOD images ultimately degrades the performance.

**Ablation on the variance value $\sigma^2$.** We show in Figure 7 (b) the effect of $\sigma^2$ — the number of the variance value for the Gaussian kernel (Section 3.2). We vary $\sigma^2 \in \{0.02, 0.03, 0.04, 0.05, 0.06, 0.2\}$. Using a mild variance value $\sigma^2$ generates meaningful synthetic OOD images for model regularization. Too large of variance (e.g., $\sigma^2 = 0.2$) produces far-OOD, which does not help learn a compact decision boundary between ID and OOD.

**Ablation on $k$ in calculating $k$-NN distance.** In Figure 7 (c), we analyze the effect of $k$, *i.e.*, the number of nearest neighbors for non-parametric sampling in the latent space. We vary $k = \{100, 200, 300, 400, 500\}$ and observe that our method is not sensitive to this hyperparameter.

**Visualization of the generated outliers.** Figure 5 illustrates the generated outlier images under different variance $\sigma^2$. Mathematically, a larger variance translates into outliers that are more deviated from ID data. We confirm this in our visualization too. The synthetic OOD images gradually become semantically different from ID classes "jellyfish" and "ladybug", as the variance increases. More visualization results are in Appendix C.

## 4.3 Extension: From DREAM-OOD to DREAM-ID

Our framework can be easily extended to generate ID data. Specifically, we can select the ID point with small $k$-NN distances *w.r.t.* the training data (Equation (5)) and sample inliers from the Gaussian kernel with small variance $\sigma^2$ in the text-conditioned embedding space (Figure 6). Then we decode the inlier embeddings via the diffusion model for ID generation (Visualization provided in Appendix G). For the synthesized ID images, we let the semantic label be the same as the anchor ID point. Here we term our extension as DREAM-ID instead.

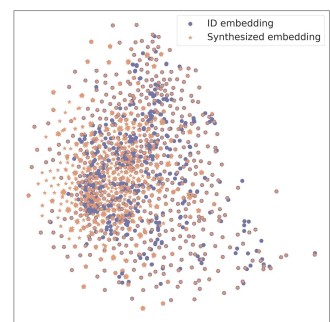

Figure 6: TSNE visualization of ID embeddings (purple) and the synthesized inlier embeddings (orange), for class "hen" in IMAGENET.

**Datasets.** We use the same IMAGENET-100 as the training data. We measure the generalization performance on both the original IMAGENET test data (for ID generalization) and variants with distribution shifts (for OOD generalization). For OOD generalization, we evaluate on (1) IMAGENET-A [44] consisting of real-world, unmodified, and naturally occurring examples that are misclassified by ResNet models; (2) IMAGENET-V2 [45], which is created from the Flickr dataset with natural distribution shifts. We provide the experimental details in Appendix H.

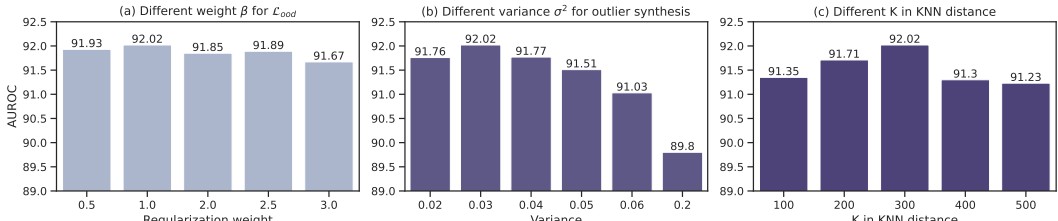

Figure 7: (a) Ablation study on the regularization weight $\beta$ on $\mathcal{L}_{ood}$. (b) Ablation on the variance $\sigma^2$ for synthesizing outliers in Section 3.2. (c) Ablation on the $k$ for the $k$-NN distance. The numbers are AUROC. The ID training dataset is IMAGENET-100.

**DREAM-ID improves the generalization performance.** As shown in Table 4, we compare DREAM-ID with competitive data augmentation and test-time adaptation methods. For a fair comparison, all the methods are trained using the same network architecture, under the same configuration. Specifically, our baselines include: the original model without any data augmentation, RandAugment [46], AutoAugment [47], CutMix [48], AugMix [49], DeepAugment [50] and MEMO [51]. These methods are shown in the literature to help improve generalization. The results demonstrate that our approach outperforms all the baselines that use data augmentation for training in both ID generalization and generalization under natural distribution shifts (↑0.74% vs. the best on IMAGENET-A, ↑0.70% vs. the best on IMAGENET-V2). Implementation details of the baselines are in Appendix I.

In addition, we compare our method with using generic prompts (*i.e.*, "A high-quality photo of a $[y]$") for data generation. For a fair comparison, we synthesize the same amount of images (*i.e.*, 1000 per class) for both methods. The result shows that DREAM-ID outperforms the baseline by $0.72\%$ on IMAGENET test set and $0.95\%, 1.20\%$ on IMAGENET-A and IMAGENET-V2, respectively.

| Methods | IMAGENET | IMAGENET-A | IMAGENET-V2 |
|---|---|---|---|
| Original (no aug) | 87.28 | 8.69 | 77.80 |
| RandAugment | 88.10 | 11.39 | 78.90 |
| AutoAugment | 88.00 | 10.85 | 79.70 |
| CutMix | 87.98 | 9.67 | 79.70 |
| AugMix | 87.74 | 10.96 | 79.20 |
| DeepAugment | 86.86 | 10.79 | 78.30 |
| MEMO | 88.00 | 10.85 | 78.60 |
| Generic Prompts | 87.74 | 11.18 | 79.20 |
| **DREAM-ID (Ours)** | **88.46**±0.1 | **12.13**±0.1 | **80.40**±0.1 |

Table 4: Model generalization performance (accuracy, in %), using IMAGENET-100 as the training data. We report standard deviations estimated across 3 runs.

## 5  Related Work

**Diffusion models** have recently seen wide success in image generation [9, 52–54], which can outperform GANs in fidelity and diversity, without training instability and mode collapse issues [55]. Recent research efforts have mostly focused on efficient sampling schedulers [14, 56, 57], architecture design [58], score-based modeling [59, 60], large-scale text-to-image generation [12, 11], diffusion personalization [61–63], and extensions to other domains [64–66].

Recent research efforts mainly utilized diffusion models for data augmentation, such as for image classification [67–75], object detection [76–78], spurious correlation [79] and medical image analysis [80–82] while we focus on synthesizing outliers for OOD detection. Graham *et.al.* [83] and Liu *et.al.* [84] utilized diffusion models for OOD detection, which applied the reconstruction error as the OOD score, and therefore is different from the discriminative approach in our paper. Liu *et.al.* [85] jointly trained a small-scale diffusion model and a classifier while regarding the interpolation between the ID data and its noisy version as outliers, which is different from the synthesis approach in DREAM-OOD. Meanwhile, our method does not require training the diffusion model at all. Kirchheim *et.al.* [86] modulated the variance in BigGAN [87] to generate outliers rather than using diffusion models. Franco *et.al.* [88] proposed denoising diffusion smoothing for certifying the robustness of OOD detection. Sehwag *et.al.* [89] modified the sampling process to guide the diffusion models towards low-density regions but simultaneously maintained the fidelity of synthetic data belonging to the ID classes. Several works employed diffusion models for anomaly segmentation on medical data [90–92], which is different from the task considered in our paper.

**OOD detection** has attracted a surge of interest in recent years [93–101]. One line of work performed OOD detection by devising scoring functions, including confidence-based methods [25, 26, 102],

energy-based score [6, 103, 104], distance-based approaches [21, 23, 27, 29, 105–108], gradient-based score [109], and Bayesian approaches [110–115]. Another promising line of work addressed OOD detection by training-time regularization [116–125]. For example, the model is regularized to produce lower confidence [4, 33] or higher energy [5, 6, 126] on the outlier data. Most regularization methods require the availability of auxiliary OOD data. [127, 128] enhanced OOD detection from the perspective of ID distribution. Several recent works [1, 2] synthesized virtual outliers in the feature space, and regularizes the model's decision boundary between ID and OOD data during training. In contrast, DREAM-OOD synthesizes photo-realistic outliers in pixel space, which enables visually inspecting and understanding synthetic outliers in a human-compatible way.

Recently, there has been growing interest in multi-modal OOD detection that utilizes textual information for visual OOD detection. Fort *et.al.* [129] proposed a scheme where pretrained CLIP models are provided with candidate OOD labels for each target dataset, and show that the output probabilities summed over the OOD labels effectively capture OOD uncertainty. Esmaeilpour *et.al.* [130] proposed to train a label generator based on the visual encoder of CLIP and use the generated labels for OOD detection. Ming *et.al.* [98, 99] alleviates the need for prior information on OOD by investigating pre-trained CLIP models and parameter-efficient fine-tuning methods for OOD detection. [131] utilized textual outlier exposure with vision-language models to enhance the neural network's capability to distinguish between ID and OOD data. In contrast, DREAM-OOD generates the outlier visual embeddings by training a classifier conditioned on the ID texts and then uses the diffusion model for outlier image synthesis.

## 6   Conclusion

In this paper, we propose a novel learning framework DREAM-OOD, which imagines photo-realistic outliers in the pixel space by way of diffusion models. DREAM-OOD mitigates the key shortcomings of training with auxiliary outlier datasets, which typically require label-intensive human intervention for data preparation. DREAM-OOD learns a text-conditioned latent space based on ID data, and then samples outliers in the low-likelihood region via the latent. We then generate outlier images by decoding the outlier embeddings with the diffusion model. The empirical result shows that training with the outlier images helps establish competitive performance on common OOD detection benchmarks. Our in-depth quantitative and qualitative ablations provide further insights on the efficacy of DREAM-OOD. We hope our work will inspire future research on automatic outlier synthesis in the pixel space.

## 7   Broader Impact

Our project aims to improve the reliability and safety of modern machine learning models. Our study on using diffusion models to synthesize outliers can lead to direct benefits and societal impacts, particularly when auxiliary outlier datasets are costly to obtain, such as in safety-critical applications. Nowadays, research on diffusion models is prevalent, which provides various promising opportunities for exploring the off-the-shelf large models for our research. Our study does not involve any violation of legal compliance. Through our study and release of code, we hope to raise stronger research and societal awareness towards the problem of data synthesis for out-of-distribution detection in real-world settings.

## 8   Acknowledgement

We thank Yifei Ming for his valuable suggestions on the draft. The authors would also like to thank NeurIPS anonymous reviewers for their helpful feedback. Research is supported by the Jane Street Graduate Research Fellowship, AFOSR Young Investigator Program under award number FA9550-23-1-0184, National Science Foundation (NSF) Award No. IIS-2237037 & IIS-2331669, Office of Naval Research under grant number N00014-23-1-2643, Philanthropic Fund from SFF, and faculty research awards/gifts from Google and Meta. Zhu is supported in part by NSF grants 2023239, 2041428, 2202457, ARO MURI W911NF2110317 and AF CoE FA9550-18-1-0166.

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

# Dream the Impossible:
## Outlier Imagination with Diffusion Models (Appendix)

## A    Details of datasets

**ImageNet-100.** We randomly sample 100 classes from IMAGENET-1K [8] to create IMAGENET-100. The dataset contains the following categories: n01498041, n01514859, n01582220, n01608432, n01616318, n01687978, n01776313, n01806567, n01833805, n01882714, n01910747, n01944390, n01985128, n02007558, n02071294, n02085620, n02114855, n02123045, n02128385, n02129165, n02129604, n02165456, n02190166, n02219486, n02226429, n02279972, n02317335, n02326432, n02342885, n02363005, n02391049, n02395406, n02403003, n02422699, n02442845, n02444819, n02480855, n02510455, n02640242, n02672831, n02687172, n02701002, n02730930, n02769748, n02782093, n02787622, n02793495, n02799071, n02802426, n02814860, n02840245, n02906734, n02948072, n02980441, n02999410, n03014705, n03028079, n03032252, n03125729, n03160309, n03179701, n03220513, n03249569, n03291819, n03384352, n03388043, n03450230, n03481172, n03594734, n03594945, n03627232, n03642806, n03649909, n03661043, n03676483, n03724870, n03733281, n03759954, n03761084, n03773504, n03804744, n03916031, n03938244, n04004767, n04026417, n04090263, n04133789, n04153751, n04296562, n04330267, n04371774, n04404412, n04465501, n04485082, n04507155, n04536866, n04579432, n04606251, n07714990, n07745940.

**OOD datasets.**   Huang *et.al.* [39] curated a diverse collection of subsets from iNaturalist [40], SUN [41], Places [36], and Texture [34] as large-scale OOD datasets for IMAGENET-1K, where the classes of the test sets do not overlap with IMAGENET-1K. We provide a brief introduction for each dataset as follows.

**iNaturalist** contains images of natural world [40]. It has 13 super-categories and 5,089 sub-categories covering plants, insects, birds, mammals, and so on. We use the subset that contains 110 plant classes which do not overlap with IMAGENET-1K.

**SUN** stands for the Scene UNderstanding Dataset [41]. SUN contains 899 categories that cover more than indoor, urban, and natural places with or without human beings appearing in them. We use the subset which contains 50 natural objects not in IMAGENET-1K.

**Places** is a large scene photographs dataset [36]. It contains photos that are labeled with scene semantic categories from three macro-classes: Indoor, Nature, and Urban. The subset we use contains 50 categories that are not present in IMAGENET-1K.

**Texture** stands for the Describable Textures Dataset [34]. It contains images of textures and abstracted patterns. As no categories overlap with IMAGENET-1K, we use the entire dataset as in [39].

**ImageNet-A** contains 7,501 images from 200 classes, which are obtained by collecting new data and keeping only those images that ResNet-50 models fail to correctly classify [44]. In our paper, we evaluate on the 41 overlapping classes with IMAGENET-100 which consist of a total of 1,852 images.

**ImageNet-v2** used in our paper is sampled to match the MTurk selection frequency distribution of the original IMAGENET validation set for each class [45]. The dataset contains 10,000 images from 1,000 classes. During testing, we evaluate on the 100 overlapping classes with a total of 1,000 images.

## B    Formulation of $Z_m(\kappa)$

The normalization factor $Z_m(\kappa)$ in Equation (3) is defined as:

$$Z_m(\kappa) = \frac{\kappa^{m/2-1}}{(2\pi)^{m/2} I_{m/2-1}(\kappa)}, \tag{8}$$

where $I_v$ is the modified Bessel function of the first kind with order $v$. $Z_m(\kappa)$ can be calculated in closed form based on $\kappa$ and the feature dimensionality $m$.

## C    Additional Visualization of the Imagined Outliers

In addition to Section 4.2, we provide additional visualizations on the imagined outliers under different variance $\sigma^2$ in Figure 8. We observe that a larger variance consistently translates into outliers that are more deviated from ID data. Using a mild variance value $\sigma^2 = 0.03$ generates both empirically (Figure 7 (b)) and visually meaningful outliers for model regularization on IMAGENET-100.

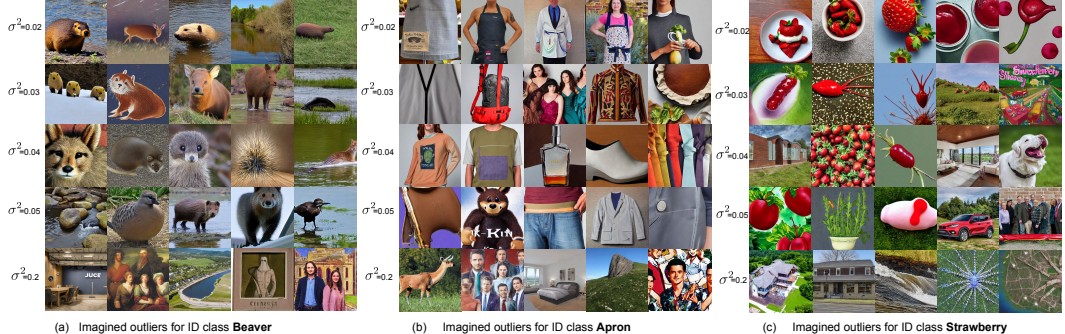

(a) Imagined outliers for ID class **Beaver**      (b) Imagined outliers for ID class **Apron**      (c) Imagined outliers for ID class **Strawberry**

Figure 8: **Visualization of the imageined outliers** for the *beaver, apron, strawberry* class with different variance values $\sigma^2$.

## D  Visualization of Outlier Generation by Embedding Interpolation

We visualize the generated outlier images by interpolating token embeddings from different classes in Figure 9. The result shows that interpolating different class token embeddings tends to generate images that are still in-distribution rather than images with semantically mixed or novel concepts, which is aligned with the observations in Liew *et. al.* [43]. Therefore, regularizing the model using such images is not effective for OOD detection (Table 2).

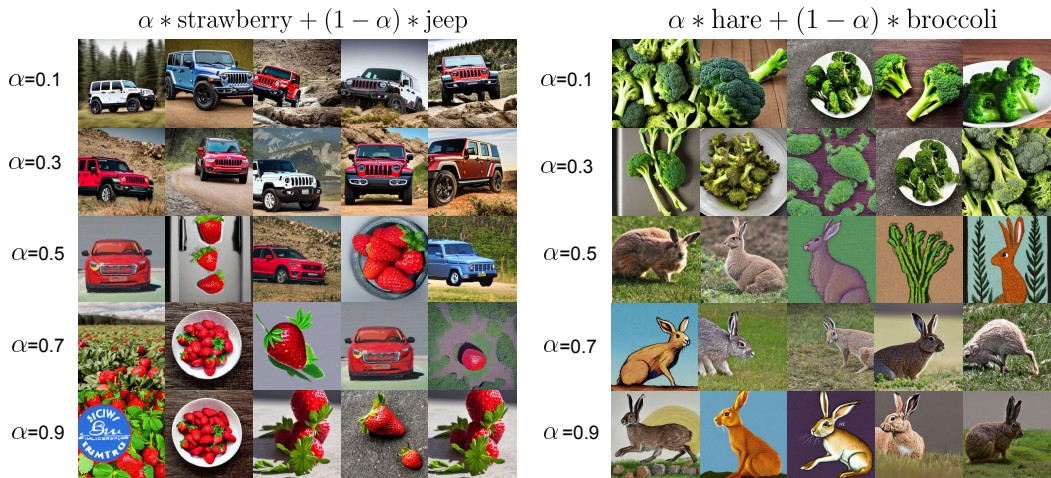

Figure 9: **Visualization of the generated outlier images** by interpolating token embeddings from different classes. We show the results with different interpolation weights $\alpha$.

## E  Visualization of the Outlier Generation by Adding Noise

As in Table 2 in the main paper, we visualize the generated outlier images by adding Gaussian and learnable noise to the token embeddings in Figure 10. We observe that adding Gaussian noise tends to generate either ID images or images that are far away from the given ID class. In addition, adding learnable noise to the token embeddings will generate images that completely deviate from the ID data. Both of them are less effective in regularizing the model's decision boundary.

## F  Comparison with Training w/ real Outlier Data.

We compare with training using real outlier data on CIFAR-100, *i.e.,* 300K Random Images [4], which contains 300K preprocessed images that do not belong to CIFAR-100 classes. The result shows that DREAM-OOD (FPR95: 40.31%, AUROC: 90.15%) can match or even outperform outlier exposure with real OOD images (FPR95: 54.32%, AUROC: 91.34%) under the same training configuration while using fewer synthetic OOD images for OOD regularization (100K in total).

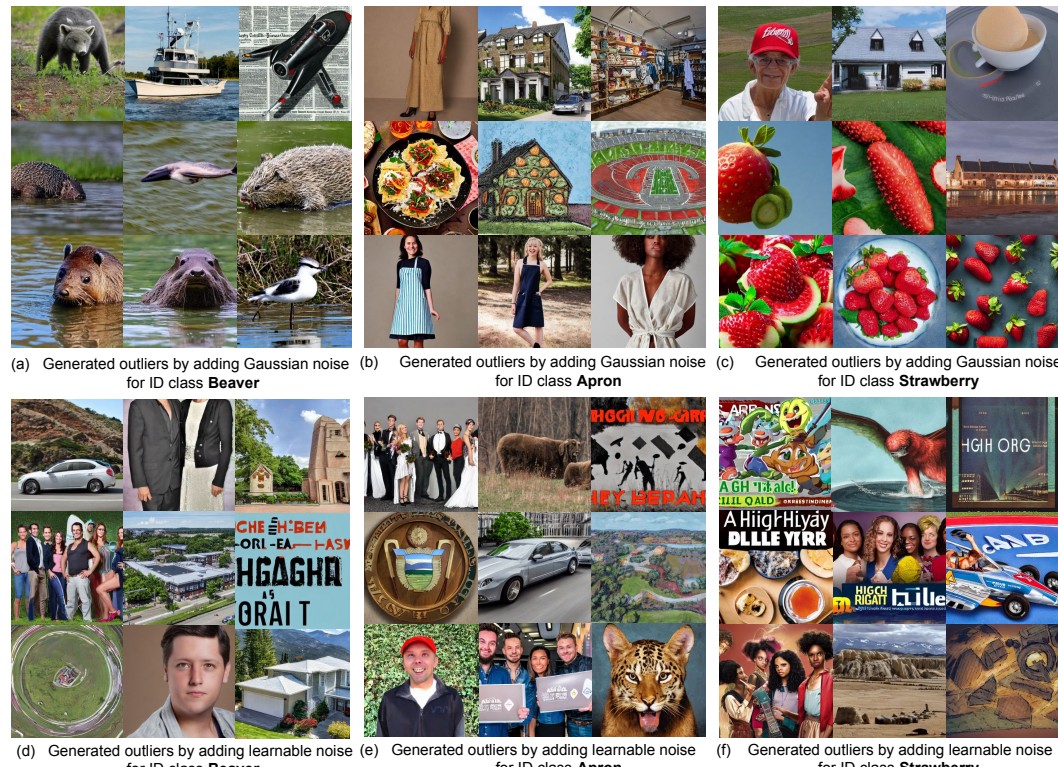

(a) Generated outliers by adding Gaussian noise for ID class **Beaver**
(b) Generated outliers by adding Gaussian noise for ID class **Apron**
(c) Generated outliers by adding Gaussian noise for ID class **Strawberry**

(d) Generated outliers by adding learnable noise for ID class **Beaver**
(e) Generated outliers by adding learnable noise for ID class **Apron**
(f) Generated outliers by adding learnable noise for ID class **Strawberry**

Figure 10: **Visualization of the generated outlier images** by adding Gaussian and learnable noise to the token embeddings from different classes.

## G   Visualization of Generated Inlier Images

We show in Figure 11 the visual comparison among the original IMAGENET images, the generated images by our DREAM-ID, and the generated ID images using generic prompts "A high-quality photo of a [cls]" where "[cls]" denotes the class name. Interestingly, we observe that the prompt-based generation produces object-centric and distributionally dissimilar images from the original dataset. In contrast, our approach DREAM-ID generates inlier images that can resemble the original ID data, which helps model generalization.

## H   Experimental Details for Model Generalization

We provide experimental details for Section 4.3 in the main paper. We use ResNet-34 [42] as the network architecture, trained with the standard cross-entropy loss. For both the CIFAR-100 and IMAGENET-100 datasets, we train the model for 100 epochs, using stochastic gradient descent with the cosine learning rate decay schedule, a momentum of 0.9, and a weight decay of $5e^{-4}$. The initial learning rate is set to 0.1 and the batch size is set to 160. We generate $1,000$ new ID samples per class using Stable Diffusion v1.4, which result in $100,000$ synthetic images. For both the baselines and our method, we train on a combination of the original IMAGENET/CIFAR samples and synthesized ones. To learn the feature encoder $h_\theta$, we set the temperature $t$ in Equation (2) to 0.1. Extensive ablations on hyperparameters $\sigma$ and $k$ are provided in Appendix J.

## I   Implementation Details of Baselines for Model Generalization

For a fair comparison, we implement all the data augmentation baselines by appending the original IMAGENET-100 dataset with the same amount of augmented images (*i.e.*, 100k) generated from different augmentation techniques. We follow the default hyperparameter setting as in their original papers.

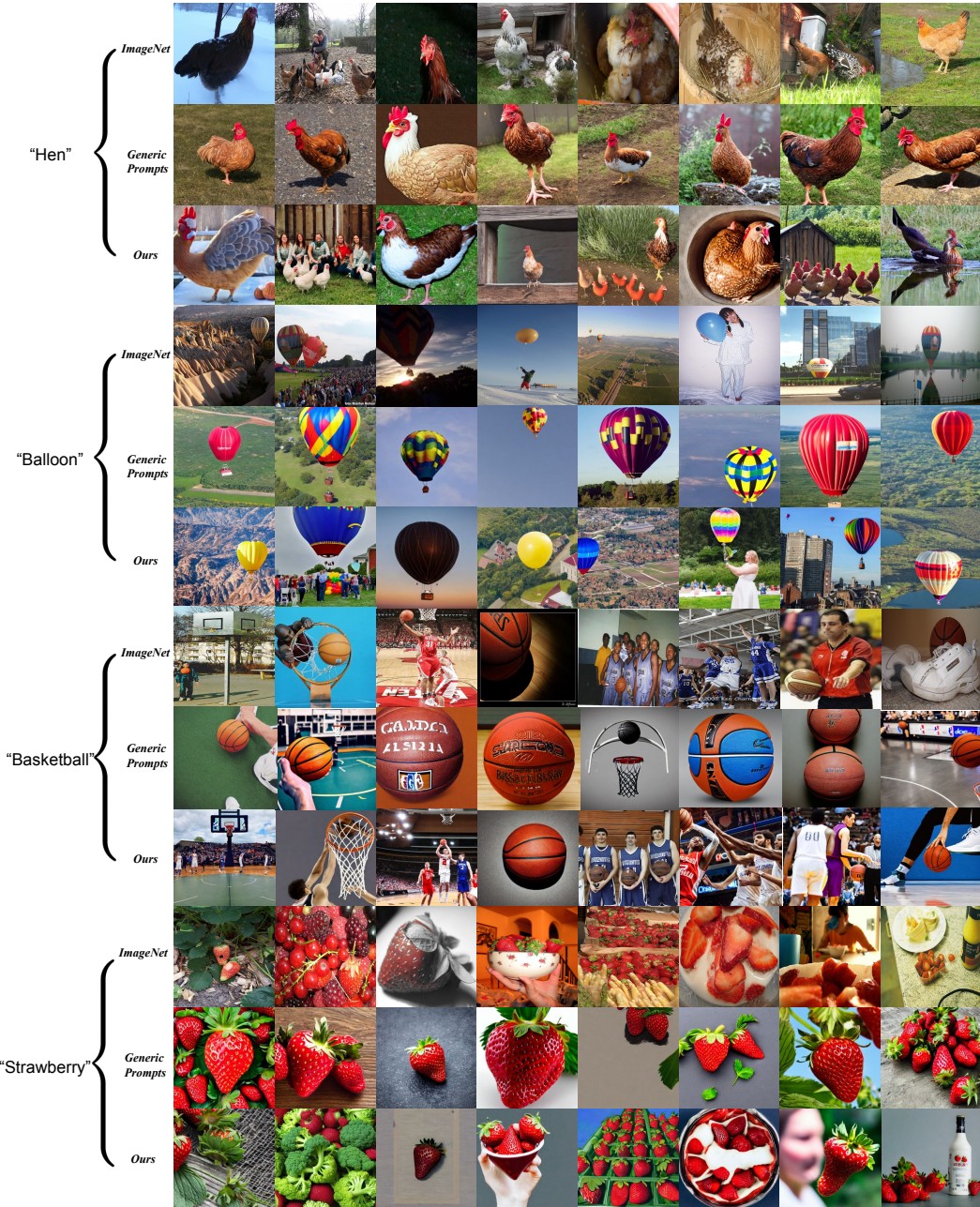

Figure 11: **Visual comparison between our DREAM-ID vs. prompt-based image generation** on four different classes.

- For RandAugment [46], we set the number of augmentation transformations to apply sequentially to 2. The magnitude for all the transformations is set to 9.

- For AutoAugment [47], we set the augmentation policy as the best one searched on IMAGENET.

- For CutMix [48], we use a CutMix probability of 1.0 and set $\beta$ in the Beta distribution to 1.0 for the label mixup.

- For AugMix [49], we randomly sample 3 augmentation chains and set $\alpha = 1$ for the Dirichlet distribution to mix the images.

- For DeepAugment [50], we directly use the corrupted images for data augmentation provided in their Github repo [3].

- For MEMO [51], we follow the original paper and use the marginal entropy objective for test-time adaptation, which disentangles two distinct self-supervised learning signals: encouraging invariant predictions across different augmentations of the test point and encouraging confidence via entropy minimization.

| Methods | IMAGENET | IMAGENET-A | IMAGENET-V2 |
|---|---|---|---|
| Original (no aug) | 87.28 | 8.69 | 77.80 |
| RandAugment | 87.56 | 11.07 | 79.20 |
| AutoAugment | 87.40 | 10.37 | 79.00 |
| CutMix | 87.64 | 11.33 | 79.70 |
| AugMix | 87.22 | 9.39 | 77.80 |
| **DREAM-ID (Ours)** | **88.46**±0.1 | **12.13**±0.1 | **80.40**±0.1 |

Table 5: **Model generalization performance (accuracy, in %), using IMAGENET-100 as the training data.** The baselines are implemented by directly applying the augmentations on IMAGENET-100.

We also provide the comparison in Table 5 with baselines that are directly trained by applying the augmentations on IMAGENET without appending the original images. The model trained with the images generated by DREAM-ID can still outperform all the baselines by a considerable margin.

# J  Ablation Studies on Model Generalization

In this section, we provide additional analysis of the hyperparameters and designs of DREAM-ID for ID generation and data augmentation. For all the ablations, we use the IMAGENET-100 dataset as the in-distribution training data.

**Ablation on the variance value $\sigma^2$.**  We show in Table 6 the effect of $\sigma^2$ — the number of the variance value for the Gaussian kernel (Section 3.2). We vary $\sigma^2 \in \{0.005, 0.01, 0.02, 0.03\}$. A small-mild variance value $\sigma^2$ is more beneficial for model generalization.

| $\sigma^2$ | IMAGENET | IMAGENET-A | IMAGENET-V2 |
|---|---|---|---|
| 0.005 | 87.62 | 11.39 | 78.50 |
| 0.01 | **88.46** | **12.13** | **80.40** |
| 0.02 | 87.72 | 10.85 | 77.70 |
| 0.03 | 87.28 | 10.91 | 78.20 |

Table 6: Ablation study on the variance value $\sigma^2$ in the Gaussian kernel for model generalization.

**Ablation on $k$ in calculating $k$-NN distance.**  In Table 7, we analyze the effect of $k$, *i.e.*, the number of nearest neighbors for non-parametric sampling in the latent space. In particular, we vary $k = \{100, 200, 300, 400, 500\}$. We observe that our method is not sensitive to this hyperparameter, as $k$ varies from 100 to 500.

| $k$ | IMAGENET | IMAGENET-A | IMAGENET-V2 |
|---|---|---|---|
| 100 | **88.51** | 12.11 | 79.92 |
| 200 | 88.35 | 12.04 | 80.01 |
| 300 | 88.46 | **12.13** | **80.40** |
| 400 | 88.43 | 12.01 | 80.12 |
| 500 | 87.72 | 11.78 | 80.29 |

Table 7: Ablation study on the $k$ for $k$-NN distance for model generalization.

---

[3] https://github.com/hendrycks/imagenet-r/blob/master/DeepAugment

## K   Computational Cost

We summarize the computational cost of DREAM-OOD and different baselines on IMAGENET-100 as follows. The post hoc OOD detection methods require training a classification model on the ID data (∼8.2 h). The outlier synthesis baselines, such as VOS (∼8.2 h), NPOS (∼8.4 h), and GAN (∼13.4 h) incorporate the training-time regularization with the synthetic outliers. Our DREAM-OOD involves learning the text-conditioned latent space (∼8.2 h), image generation with diffusion models (∼10.1 h for 100K images), and training with the generated outliers (∼8.5 h).

## L   Software and hardware

We run all experiments with Python 3.8.5 and PyTorch 1.13.1, using NVIDIA GeForce RTX 2080Ti GPUs.

