# OpenReview forum: "Dream the Impossible: Outlier Imagination with Diffusion Models"
_NeurIPS.cc/2023/Conference — NeurIPS 2023 poster_

### Official Review · Reviewer_i54j · 2023-07-02

**Soundness:** 2 fair
**Presentation:** 3 good
**Contribution:** 2 fair
**Rating:** 5
**Confidence:** 3

**Summary:**

The paper proposes a learning framework to generate outliers in the pixel space by way of diffusion models with only the in-distribution data. By learning a text-conditioned latent space based on  in-distribution data, the methods further sample outlier latents in low-likelihood regions. The empirical result shows that training with the generated outlier images helps establish competitive performance on common OOD detection benchmarks.

**Strengths:**

- The paper is well-written and clear to me.
- The ablation studies are sufficient.

**Weaknesses:**

Please refer to the questions.

**Questions:**

- The OOD sampling methods are proposed in [1]. [1] uses the sampled latent for direct training while the method in this paper uses diffusion for further training. The key idea is incremental.
- About the generator: The proposed method generates OOD samples based on a powerful stable diffusion model, which is trained on a large-scale dataset. It is much easier to generate unseen-class images. Is there any possibility for data leakage? The prior method[2] uses the diffusion trained on the original dataset. Does improvement over[1] come from the strong diffusion model?
- How is the feature encoder designed? Have you tried a pre-trained CLIP image encoder for initialization? Considering that  the ability of the feature encoder has a direct impact on the sampling of OOD latent.
- The cifar10 and imagenet are also commonly-used benchmarks. Besides cifar100 and imagenet100,  I wonder if there are any other results for these two more challenging datasets.

[1] Tao, L., Du, X., Zhu, X. and Li, Y., 2023. Non-parametric outlier synthesis.

[2] Liu, L., Ren, Y., Cheng, X. and Zhao, Z., 2022. Diffusion denoising process for perceptron bias in out-of-distribution detection.

**Limitations:**

The authors have adequately addressed the broader impacts.

---

> ### Author Rebuttal · Authors · 2023-08-08
>
> We thank the reviewer for the thorough comments, which we address below:
>
> **A1. Clarification on our contribution**
>
> We summarize the non-trivial differences between NPOS and DREAM-OOD below:
>
> - First, our key contribution is to enable the generation of high-resolution outliers for OOD detection, which was impossible with NPOS. DREAM-OOD advances the field by helping researchers visualize and understand the imagined outliers in the pixel space, **which not only offers better empirical performance but also much more interpretability on the generated outliers**. We believe this is a meaningful scientific contribution w.r.t. prior work.
> - Second, our paper contributes non-trivial technical solutions to enable using a diffusion model for outlier synthesis. Naively taking the outliers generated by NPOS won't work for the diffusion model, because NPOS generates visual latent embedding whereas the diffusion model expects a textual latent as input. While the idea of DREAM-OOD may seem natural in hindsight, it wasn't obvious to us how to generate meaningful text-latent-based outliers that one can not only use for the diffusion model but also effectively regularize the classifier. To resolve this challenge, a key intellectual design of DREAM-OOD as you recognized, is to learn a text-conditioned latent space for sampling OOD embeddings, so that the visual embeddings are aligned with the corresponding textual embeddings. Moreover, looking at the overall method design, DREAM-OOD is much more challenging than NPOS, since it encapsulates seamlessly multiple key components including learning text-conditioned latent space, sampling, decoding, and finally learning with the imagined outliers. The entire intellectual and experimental effort to make things work wasn't trivial :)
> - Lastly, our method can be used to generate ID data and improves the model generalization performance, which is not explored in NPOS. We show stronger performance than the most popular data augmentation methods, which will benefit an even broader community.
>
> We believe the major differences support the novel contributions of our approach. Meanwhile, the novelty of our framework with diffusion models is recognized by other reviewers, such as:
>
> > 1) "_The paper is the first to generate high-resolution outliers for OOD detection task. It is a novel use of diffusion models_." from reviewer 3rTL
> > 2) "_This paper proposes a new framework...which is the first method for generating photo-realistic outliers and owns high originality_." from reviewer 94Wq
>
> **A2. Discussion on data leakage with diffusion models**
>
> The recent rise of large-scale generative models has enabled thousands of research projects, such as subject-driven generation [1], open-vocabulary segmentation [2], etc. Data leakage is common in all research projects that are built on modern diffusion models. Ours perhaps is not an exception. Although this is outside the focus of our current research scope, we do believe it is an important issue for the research community.
>
> To mitigate the possible data leakage concern, one can potentially replace the diffusion model to be trained from scratch. In this vein, [3] provides a promising route. We are happy to incorporate your opinion in the revised draft with proper citations including [3].
>
> [1] Ruiz et.al., DreamBooth: Fine Tuning Text-to-Image Diffusion Models for Subject-Driven Generation, CVPR 2023.
>
> [2] Xu et.al., Open-vocabulary panoptic segmentation with text-to-image diffusion models, CVPR 2023.
>
> [3] Liu et.al., Diffusion denoising process for perceptron bias in out-of-distribution detection, arXiv preprint 2211.11255.
>
> **A3. Discussion on the improvement over NPOS**
>
> Compared with NPOS, DREAM-OOD can generate high-resolution outliers in the pixel space. The higher-dimensional pixel space offers much more knowledge about the unknowns, which provides the model with high variability and fine-grained details for the unknowns that are missing in NPOS. Since our method is more photo-realistic and better for humans, the generated images can be naturally better constrained for NNs (for example, things may be more on the natural image manifolds) – that’s not available in NPOS. As shown in Figure 5, the generated outliers are more precise in characterizing OOD data and thus improve the empirical performance.
>
> **A4. Experiments on using CLIP for feature encoder**
>
> We use a ResNet architecture for the feature encoder. As suggested, we tried fine-tuning the CLIP visual encoder (ViT-L/14), while keeping other parts unchanged. The comparison is shown as follows, where using CLIP as the initialized feature encoder achieves similar performance as DREAM-OOD.
>
> | | INATURALIST | |PLACES ||SUN||TEXTURES||Average ||ID ACC|
> | ------ | ----- | ----- |  ----- | ----- |----- | ----- |----- | -----|----- |----- |----- |
> | Method | FPR95 | AUROC | FPR95 | AUROC | FPR95 | AUROC |FPR95 | AUROC | FPR |  AUROC ||
>  |DREAM-OOD   |  24.10 | 96.10 | **39.87** | 93.11 | **36.88** | 93.31 | 53.99 | 85.56 | **38.76** | 92.02 | 87.54|
> | CLIP as initialized latent |  **21.60**| **96.36**|43.82 | **93.20** | 38.00| **94.58**| **52.94**| **87.64**|39.09 | **92.94**| 87.06|
>
> **A5. Results on additional datasets**
>
> As suggested, we compare DREAM-OOD with the baselines on an additional CIFAR-10 dataset. We use ResNet-18 for training. Other training details are kept the same as CIFAR-100. The result is shown in **Table 1 in the [PDF](https://openreview.net/attachment?id=vmDoMZMncP&name=pdf)**, where the strong performance of DREAM-OOD still holds.
>
> Due to the synthesis task being more challenging, literature including NPOS has primarily focused on ImageNet-100, which we closely follow and adopt as well. Given ImageNet-100 consists of high-resolution images, we believe the results and improvement over NPOS can already meaningfully support the benefits of DREAM-OOD on real-world datasets. We plan to include the full ImageNet-1k results in our revision.

---

> > ### Comment · Reviewer_i54j · 2023-08-20
> > **Thanks for the response**
> >
> > Thanks for the responses. Most of questions have been resolved. Even if I still concern about the data leakage and it will be much more meaningful to find the OOD samples only based on the original datasets without extra data, I  still admire the exploration and will raise my score to slightly positive side.

---

> > > ### Author Response · Authors · 2023-08-20
> > > **thanks for your follow-up**
> > >
> > > We are glad to hear that our response helped resolved your questions! We thank the reviewer for taking the time to read our rebuttal and for your positive feedback again!
> > >
> > > Best,
> > >
> > > Authors

---

> ### Author Response · Authors · 2023-08-17
> **Thanks for your feedback**
>
> Dear Reviewer i54j,
>
> As the discussion period ends soon, we just wanted to check if the response clarified your questions. Thanks again for your constructive feedback.
>
> Best,
>
> Authors

---

### Official Review · Reviewer_3rTL · 2023-07-04

**Soundness:** 3 good
**Presentation:** 3 good
**Contribution:** 2 fair
**Rating:** 6
**Confidence:** 3

**Summary:**

The paper tackles OOD detection in the image space and proposes to generate outliers with diffusion models. Specifically, the authors find embeddings in the text-conditioned latent space that are on the boundary of in-distribution embedding clusters. These embeddings are likely to be those of strong outliers. By performing denoising with diffusion models, outlier images can then be generated and used for OOD classifier training. Empirical results show that the approaches perform better than many baselines.

**Strengths:**

* The paper is the first to generate high-resolution outliers for OOD detection task. It is a novel use of diffusion models.
* The paper is well-written with clear method and experiment presentation.

**Weaknesses:**

* It seems that by adding simple Gaussian noise to the text label embedding space instead of training a classifier can already be quite strong. Is the classifier necessary here? Nevertheless, I appreciate the detailed ablation on this.
* Please see Questions section for additional comments.

**Questions:**

1. Can the authors provide possible explanations on why the proposed approach performs consistently worse than baselines on the texture OOD dataset? Would some visualization help explain what happened?
1. There exists an unsupervised OOD detection setting where the labels of in domain data are not provided. How would the proposed approach be extended in this setting?
1. The authors might also consider using negative prompts to find additional outliers. That is, instead of specifying what the outlier should look like, it might be possible to avoid the class by pushing the embedding away from that class. Currently, all outliers are somewhat associated with in-domain data classes. It would be ideal to be able to generate outliers from additional classes.


**Limitations:**

No limitations mentioned in the main text.

---

> ### Author Rebuttal · Authors · 2023-08-08
>
> We are encouraged that the reviewer found our approach novel and the paper well-written. We address comments in detail below:
>
> **A1. Clarification on adding simple Gaussian noise to the latent space**
>
> Thanks for acknowledging our ablations on different outlier image synthesis methods!  We agree with your opinion that adding the Gaussian noise to the textual embeddings is a strong synthesis baseline.
>
> In contrast to pure noise, our framework DREAM-OOD learns a text-conditioned latent space for outlier generation, which can *sample low-likelihood embeddings based on the learned feature representations* and is more principled than adding Gaussian noise. The rationale of such a design is that if the sampled embeddings are distributionally far away from the ID embeddings, the decoded images will have a large semantic discrepancy with the ID images and vice versa. Adding Gaussian noise does not necessarily guarantee that the sampled outliers are in the low-likelihood region. Therefore, our approach is theoretically more sound and leads to **consistently better** OOD detection performance than adding noise to the textual embeddings across different ID datasets.
>
> **A2.Explanations of the worse performance on the texture OOD dataset**
>
> During the rebuttal, we examined the penultimate-layer features for the ID data and the OOD Textures dataset. We found that after regularizing the neural network using the generated outlier images, the ID and OOD features are somewhat overlapped.
>
> We hypothesize the reason might be that the generated outlier images usually have diverse backgrounds/objects, either with indoor or outdoor environments/objects. When regularizing the classification model using such images, the model might be more confident in recognizing the images with similar background/objects as OOD while the real OOD texture images deviate from such synthesized outlier image distribution, which leads to inferior OOD detection performance.
>
> **A3. Discussion on the extension in the unsupervised OOD detection setting**
>
> Since the vast majority of OOD detection literature focuses on the supervised setting (see related work L321-L331), DREAM-OOD considers the ID labels. Without ID semantic labels provided, one simple extension to the unsupervised setting can be (1) firstly generating the captions for the unlabeled ID dataset using a state-of-the-art image-captioning model, such as [1]; (2) Collecting the object names in the generated captions; (3) Learning the text-conditioned latent space and then synthesizing the OOD outlier images as in DREAM-OOD.
>
>
> [1] Li et.al., BLIP-2: Bootstrapping Language-Image Pre-training with Frozen Image Encoders and Large Language Models, ICML 2023.
>
> **A4. Additional design choices of the OOD embedding synthesis**
>
> You raised a great point! To respond to your suggestions:
>
> - Your idea of pushing the embedding away from that class coincides with the baseline *(II) Add learnable noise* in Table 2 of the submission :) In this baseline, we add learnable noise to the token embeddings where the noise is trained to push the outliers away from ID features. Though adding noise to the token embedding is relatively simple, it cannot explicitly sample textual embeddings from the low-likelihood region as DREAM-OOD does, which are near the ID boundary and thus demonstrate stronger effectiveness to regularize the model. For your reference, prior work [2] showed that near-boundary outliers are the most informative for OOD detection since they help learn a decision boundary that's tightly surrounding ID data.
> - As for the idea of generating outliers from additional classes, we have implemented it during rebuttal and compared it with our DREAM-OOD. Specifically, we use the remaining 900 classes in ImageNet-1k (exclude the 100 classes in ImageNet-100 and CIFAR-100) as the disjoint class names for outlier generation. We generate the same amount of images as our DREAM-OOD to ensure a fair comparison. The OOD detection results are summarized in the following table, where we observe worse performance. We hypothesize that the generated outlier images are relatively far away from the decision boundary between the in-distribution and OOD.
>
> | Method | FPR95 | AUROC | FPR95 | AUROC |
> | ------ | ----- | ----- |  ----- | ----- |
> |  | ImageNet-100 | |CIFAR-100 |
>  |DREAM-OOD   |  **38.76**| **92.02** |**40.31** |**90.15** |
> | Using other classes  | 43.55 | 87.84| 49.89| 85.87 |
>
> [2] Ming et.al., POEM: Out-of-Distribution Detection with Posterior Sampling, ICML 2022.

---

> > ### Comment · Reviewer_3rTL · 2023-08-17
> > **Thank you for the response**
> >
> > I acknowledge I have read the author response and other reviews. The answers are generally useful and I am keeping my score unchanged.

---

> > > ### Author Response · Authors · 2023-08-17
> > > **thank you**
> > >
> > > We would like to thank the reviewer for taking the time to read our rebuttal and for your positive feedback again!
> > >
> > > Best,
> > > Authors

---

### Official Review · Reviewer_thbj · 2023-07-04

**Soundness:** 3 good
**Presentation:** 3 good
**Contribution:** 3 good
**Rating:** 7
**Confidence:** 2

**Summary:**

This paper proposes DREAM-OOD.
It constructs a text-conditioned latent space by learning an image encoder $h_\theta$ with a pretrained text encoder $\mathcal{T}$ and a contrastive loss.
During OOD generation, DREAM-OOD first generates outliers in the latent space according to the k-NN distance in Eq. (5), and then decodes them to the pixel space using a pretrained diffusion model.
DREAM-OOD can also be extended to ID data synthesis.
Experiments show that DREAM-OOD provides benefits for OOD detection tasks.

**Strengths:**

1. This paper is well-organized, and the method is easy to follow.
2. The t-SNE visualizations in Figure 3 and 4 clearly demonstrate the quality of the learned latent space and the feasibility of filtering latent outliers using Eq. (5).
3. DREAM-OOD outperforms selected baselines using CIFAR-100 or ImageNet as the ID dataset.
4. This paper includes ablation studies on the selection of some hyperparameters. Figure 7 shows that DREAM-OOD is robust to these hyperparameters.

**Weaknesses:**

1. There is neither a complexity analysis nor a report on the computational resource cost.
2. Although ablation studies on hyperparameters are included, the range of the candidates is still problematic. For example, the range of $k$ is too small compared to the dataset size.

**Questions:**

1. How does DREAM-OOD scale to larger datasets? The ID datasets used in this paper are quite small.

**Limitations:**

There is no discussion on the limitation.

---

> ### Author Rebuttal · Authors · 2023-08-08
>
> We are happy to see that the reviewer finds our work easy to follow with appropriate visualizations & ablations and that our paper is organized. We thank the reviewer for the thorough comments and suggestions, which we address below:
>
>
> **A1.Discussion on the computation cost**
>
> As suggested, we summarize the computational cost of DREAM-OOD and different baselines on ImageNet as follows.
>
>
> Specifically, DREAM-OOD involves learning the text-conditioned latent space ($\sim$ 8.2 h), image generation with diffusion models ($\sim$ 10.1 h for 100K images), and training with the generated outlier images ($\sim$ 8.5 h). We use 8 NVIDIA GeForce RTX 2080Ti GPUs for our experiments.
>
> We have included the computational cost in the revised manuscript.
>
> **A2. Discussion on the range of k in the ablation**
>
> Thanks for your suggestion! Since we calculate the $k$-NN distance between each sample and the remaining samples belonging to the same class in implementation, and there are approximately 1,000 samples **per class** for the ImageNet dataset and 500 samples per class for the CIFAR-100 dataset, we set the limit of $k$ to 500. For the ImageNet dataset, we additionally experimented with larger values below, with $k=600,700,800,900,1000$, which shows less superior OOD detection results.
>
> | $k$  | AUROC |
> | ------ | ----- |
> | 300 (best) | **92.02**|
> | 600   | 91.34|
> | 700   | 91.06  |
> | 800   |  90.08 |
> | 900   |  89.18 |
> | 1000   |  88.27 |
>
> **A3. Scalability of DREAM-OOD on larger ID datasets**
>
> Great point! The largest ID dataset we used in the experiments contains approximately 100K images. For larger ID datasets, DREAM-OOD scales well for the steps of learning the text-conditioned latent space and learning with imagined outlier images. For generating the OOD images with the Stable Diffusion model, the cascaded diffusion backward sampling procedure will take longer time as they generally need more sequential function evaluation steps of large neural networks.
>
> Therefore, *the scalability of DREAM-OOD essentially hinges on the scalability of large denoising diffusion models*. In practice, we have implemented our algorithm using multiple optimized engineering techniques, such as fast sampling schedulers (i.e., PLMS [1]) with only 50 backward iterations. We have also implemented the Flash Attention [2] for the Stable Diffusion model to speed up image generation. As a next step, we will be very interested in testing DREAM-OOD on larger datasets, such as ImageNet-1k as ID, which is  10 times larger than the ID datasets used in our paper.
>
> [1] Liu et.al., Pseudo Numerical Methods for Diffusion Models on Manifolds, ICLR 2022.
>
> [2] Dao et.al., Flashattention: Fast and memory-efficient exact attention with io-awareness, NeurIPS 2022.

---

> > ### Comment · Reviewer_thbj · 2023-08-17
> > **Acknowledgement**
> >
> > Thanks for your response to address my questions.

---

> > > ### Author Response · Authors · 2023-08-17
> > > **Thanks for your response**
> > >
> > > Thank you for taking the time to read our rebuttal and your constructive feedback again!
> > >
> > > Best,
> > >
> > > Authors

---

### Official Review · Reviewer_94Wq · 2023-07-05

**Soundness:** 3 good
**Presentation:** 4 excellent
**Contribution:** 3 good
**Rating:** 7
**Confidence:** 3

**Summary:**

This paper focuses on utilizing auxiliary outliers for the OOD detection task. Specifically, different from other works using the collected outliers, this work studies generating photo-realistic outliers in the high dimensional pixel space. This paper proposes a new framework, namely, DREAM-OOD, which utilizes diffusion models to generate outliers with only in-distribution data and classes. Comprehensive analyses and experiments are conducted to demonstrate the effectiveness of the proposed method.

**Strengths:**

1. This paper proposes a new framework, i.e., DREAM-OOD, for image synthesis, which is the first method for generating photo-realistic high-resolution outliers and owns high originality.
2. The outlier imagination with diffusion models utilizes the text-conditioned latent space and has the theoretical interpretation of the loss, the presentation is good and easy to understand.
3. The experimental parts include analyses of different perspectives, and the performance of DREAM-OOD is promising.

**Weaknesses:**

1. Apart from involving the diffusion model, what is the difference in technical level between the previous methods (like VOS and NPOS) on outlier synthetics with the proposed method in image synthetic?
2. This method may be affected by the quality of the pre-trained diffusion models. What if the diffusion model can not generate high-resolution images?



**Questions:**

1. Apart from involving the diffusion model, what is the difference in technical level between the previous methods (like VOS and NPOS) on outlier synthetics with the proposed method in image synthetic?
2. This method may be affected by the quality of the pre-trained diffusion models. What if the diffusion model can not generate high-resolution images?

**Limitations:**

There is no discussion of the limitations of the current work.

---

> ### Author Rebuttal · Authors · 2023-08-08
>
> We are glad that the reviewer finds our work new, owns high originality, and is easy to understand, with promising results and analyses. We thank the reviewer for the thorough comments and suggestions, which we address below:
>
> **A1.Technical differences between DREAM-OOD and related work**
>
> Great point! Technically, there are two main differences between DREAM-OOD and the mentioned related works, such as VOS and NPOS.
>
> - First, DREAM-OOD introduces a crucial step to align the in-distribution visual embeddings with the corresponding textual embeddings from the diffusion model (see Section 3.1). Learning the text-conditioned latent space is the key to leveraging diffusion-based models for decoding and generating photo-realistic outliers in the pixel space. In contrast, VOS and NPOS do not leverage diffusion models, and directly sample outliers in the feature space of the image classifier.
> - Secondly, DREAM-OOD decodes the sampled OOD embeddings via the diffusion models to obtain the outlier images in the pixel space, which are then used for model regularization. Instead, VOS and NPOS directly separate the outliers and the ID in the feature space.
> - Thirdly, DREAM-OOD can also be extended to augment ID data distribution, enriching the data diversity for better generalization (Section 4.3). This was not demonstrated in either VOS or NPOS.
>
> **A2. Discussion on the quality of the pre-trained diffusion models**
>
> Another great point! The landscape of AI has been rapidly changing in the last year. In particular, the recent rise of large-scale generative models has provided researchers in the community with many powerful diffusion models that can produce high-resolution images. The wide availability of strong diffusion models is a key reason why DREAM-OOD becomes possible (which admittedly, would have been difficult in the past). Notably, there are more than thousands of research projects built on top of them, such as subject-driven generation [1], open-vocabulary segmentation [2], spurious correlation [3], etc., to name a few. In a similar spirit, DREAM-OOD explores the promise of diffusion models for improving OOD detection, which shows strong results both quantitatively and qualitatively. We will discuss the reliance of our approach on the diffusion models in the revised version.
>
>
> [1] Ruiz et.al., DreamBooth: Fine Tuning Text-to-Image Diffusion Models for Subject-Driven Generation, CVPR 2023.
>
> [2] Xu et.al., Open-vocabulary panoptic segmentation with text-to-image diffusion models, CVPR 2023
>
> [3] Jain et.al., Distilling model failures as directions in latent space, ICLR 2023.

---

> > ### Comment · Reviewer_94Wq · 2023-08-17
> > **Thanks for the detailed response!**
> >
> > Thanks for the detailed response! The detailed and insightful highlights well address the previous concerns. The reviewer appreciates the authors' efforts in clarifying the technical differences. It shows the uniqueness of DREAM-OOD to not only generate photo-realistic outliers in the pixel space but also can augment the ID distribution for better generalization. It would be better if the authors could also discuss the potential of DREAM-OOD to enhance OOD detection from the perspective of ID distribution, as some recent studies [1,2] also pointed out the importance of ID data quality.
> >
> > [1] In or Out? Fixing ImageNet Out-of-Distribution Detection Evaluation. ICML 2023
> > [2] Unleashing Mask: Explore the Intrinsic Out-of-Distribution Detection Capability. ICML 2023

---

> > > ### Author Response · Authors · 2023-08-17
> > > **Thanks for your additional feedback**
> > >
> > > Thank you for taking the time for reading our rebuttal!
> > >
> > > For your additional comments, we think it would be useful to extend the DREAM-OOD to 1) generate diverse ID samples for improved ID classification and 2) near-boundary ID samples for binary classification between ID and OOD. For the former extension, [1] suggests there might be a close relationship between the OOD detection performance and the ID classification accuracy. Therefore, a better ID accuracy might lead to a better OOD detection result. For the latter extension, generating the near-boundary ID samples will help the model learn an even better decision boundary between ID and OOD, which is shown to benefit OOD detection [2].
> > >
> > > We will be happy to incorporate the discussion in the revised version with proper citations to the mentioned papers.
> > >
> > > [1] Vaze et.al., Open-Set Recognition: a Good Closed-Set Classifier is All You Need?, ICLR 2022
> > >
> > > [2] Ming et.al., POEM: Out-of-Distribution Detection with Posterior Sampling, ICML 2022.

---

> > > > ### Comment · Reviewer_94Wq · 2023-08-17
> > > > **Thanks for the further reply!**
> > > >
> > > > Thanks for the further discussion! I have no further concerns and would like to increase my score for the recommendation. Thanks!

---

> > > > > ### Author Response · Authors · 2023-08-18
> > > > > **thank you**
> > > > >
> > > > > We really appreciate the support here - thank you!

---

### Official Review · Reviewer_Qkax · 2023-07-06

**Soundness:** 3 good
**Presentation:** 3 good
**Contribution:** 3 good
**Rating:** 5
**Confidence:** 3

**Summary:**

This paper proposes DREAM-OOD, a new diffusion-models-based framework to enable the generation of photo-realistic high-resolution outliers for OOD detection. DREAM-OOD works by training a text-conditioned latent space using ID data and then samples the outliers in the low-likelihood region of the latent space. Empirical results demonstrate the effectiveness of the proposed framework in enhancing the performance of OOD detection.

**Strengths:**

1. The paper is well-written with clear and detailed explanations.
2. The paper proposed an effective and reasonable framework for generating high-quality outliers images for OOD detection and achieves good empirical performance.
3. Comprehensive and extensive experiments were conducted to compare the performance of DREAM-OOD to other existing OOD detection models and provide deep insights on the efficacy of the proposed method.

**Weaknesses:**

1. The paper only emphasizes that the proposed method can allow us to understand the outliers in a human-compatible way compared to other synthetic methods such as VOS and NPOS, but it is not very clear why the proposed method can achieve better OOD detection compared to others. And there is no related theoretical analysis.
2. It does not mention the computational cost when comparing the proposed method with other OOD detection methods.
3. It will be better if the related works can be placed before section 3.
4. The novelty of the paper is not strong enough since it is not the first work to leverage diffusion models or visual latent space to generate outliers to promote  OOD detection.

**Questions:**

1. The paper suggests learning a text-conditioned latent space based on ID data, which is task specific. I wonder how the method is superior to directly leveraging the visual latent space of the pre-trained vision-and-language model CLIP.
2. What is the three-layer nonlinear MLP function φ(·) used for in equation (7)? Is it the binary logistic classifier for OOD detection?

**Limitations:**

The authors have not addressed the limitations and potential negative social impacts of their work. And I do not see any obvious negative social impacts specific to this work.

---

> ### Author Rebuttal · Authors · 2023-08-08
>
> We are encouraged that you recognize our method to be new, effective, and reasonable, and with comprehensive and extensive empirical results. We thank the reviewer for the thorough comments and suggestions, which we address below:
>
> **A1. Discussion on DREAM-OOD versus feature-based outlier synthesis**
>
> Great point! Compared with the feature-based outlier synthesis approaches, DREAM-OOD enables generating photo-realistic high-resolution outliers in the pixel space. The higher-dimensional pixel space can offer much more knowledge about the unknowns, which provides the neural networks with high variability and fine-grained details for the unknowns that are missing in the feature-based synthesized outliers  (such as VOS and NPOS). Since our method is more photo-realistic and better for humans; the generated images can be naturally better constrained for NNs -- that's not available in old methods (for example, things may be more on the natural image manifolds). As shown in Figure 5, the generated outlier images demonstrate reasonably creative OOD visual representations, which are more precise in characterizing OOD data and thus improve performance.
>
> Moreover, we would like to clarify that our paper focuses on *demonstrating the feasibility and promise of generating photo-realistic outliers in the high dimensional pixel space by way of diffusion models*. We provide comprehensive quantitative and qualitative analyses to understand the efficacy of DREAM-OOD. While we position DREAM-OOD as a methodology rather than a theory paper, we do agree it will be a meaningful direction. One plausible direction can be analyzing the generalization error bound for the OOD detector [1] trained with the ID and the generated outlier images. One can use the probably approximately correct (PAC) learning theory and analyze the key concepts, such as sample complexity, in the setting of learning with the auxiliary outliers and compare our approach against the feature-based synthesis approaches. We believe our current work serves as a key cornerstone to enable such theoretical analysis next.
>
> [1] Fang et.al., Is Out-of-Distribution Detection Learnable?, NeurIPS 2022.
>
> **A2. Discussion on the computational cost**
>
> As suggested, we summarize the computational cost of DREAM-OOD and different baselines on ImageNet as follows.
>
>
> Specifically, the post hoc OOD detection methods require training a classification model on the ID data ($\sim$ 8.2 h). The outlier synthesis baselines, such as VOS ($\sim$ 8.2 h), NPOS ($\sim$ 8.4 h), and GAN ($\sim$ 13.4 h) incorporate the training-time regularization with the synthetic outliers. Our DREAM-OOD involves learning the text-conditioned latent space ($\sim$ 8.2 h), image generation with diffusion models ($\sim$ 10.1 h for 100K images), and training with the generated outliers ($\sim$ 8.5 h). We use 8 NVIDIA GeForce RTX 2080Ti GPUs for our experiments.
>
> We have included the computational cost in the revised manuscript.
>
> **A3. Novelty**
>
> To the best of our knowledge, DREAM-OOD is the first work to leverage diffusion models for synthesizing high-quality outliers in the pixel space. We conducted an extensive literature survey (see L307-320), where no existing work is similar to ours in terms of scope and methodology. While Graham et.al. [2] and Liu et.al. [3] utilized diffusion models for OOD detection, they directly applied the reconstruction error as the OOD score and did not focus on outlier synthesis. In contrast to the feature-based outlier synthesis methods, DREAM-OOD enables the new capability of synthesizing in the high-dimensional pixel space, which not only offers better empirical performance but also much more visual interpretability on the generated outliers.
>
> We believe the major differences w.r.t. literature support the meaningful contributions of our proposed approach. At the same time, the novelty of our learning framework is recognized by several other reviewers, for example:
>
> > 1) "_The paper is the first to generate high-resolution outliers for OOD detection task. It is a novel use of diffusion models_." from reviewer 3rTL
> > 2) "_This paper proposes a new framework, i.e., DREAM-OOD, for image synthesis, which is the first method for generating photo-realistic high-resolution outliers and owns high originality_." from reviewer 94Wq
>
>
> [2] Graham et.al., Denoising diffusion models for out-of-distribution detection, CVPR VAND Workshop 2023
>
> [3] Liu et.al., Diffusion denoising process for perceptron bias in out-of-distribution detection, arXiv preprint 2211.11255.
>
> **A4. Experiments on the text-conditioned latent space versus the visual latent space of CLIP**
>
> That's an interesting idea. As suggested, we have implemented sampling by directly using the latent space of the pre-trained CLIP model (ViT-L/14 as the backbone). All other settings are kept the same as the DREAM-OOD. The results comparison on ImageNet is shown below:
>
> | | INATURALIST | |PLACES ||SUN||TEXTURES||Average ||ID ACC|
> | ------ | ----- | ----- |  ----- | ----- |----- | ----- |----- | -----|----- |----- |----- |
> | Method | FPR95 | AUROC | FPR95 | AUROC | FPR95 | AUROC |FPR95 | AUROC | FPR |  AUROC ||
>  |DREAM-OOD   |  **24.10** | **96.10** | **39.87** | **93.11** | **36.88** | **93.31** | **53.99** | **85.56** | **38.76** | **92.02** | 87.54|
> | CLIP as latent |47.28  | 88.80| 47.23|84.22  |39.09 |85.82 | 68.75| 79.69| 50.57|84.63 | 86.92 |
>
> As we can observe, using the pre-trained CLIP latent for sampling achieves a worse performance compared to our text-conditioned latent space because the pre-trained CLIP latent is suboptimal compared to ours.
>
> **A5. Clarification on the three-layer nonlinear MLP function**
>
> The three-layer nonlinear MLP function is the binary logistic classifier for OOD detection as you concur.
>
> **A6. Position of the related work section**
>
> We will rearrange the sections based on your suggestion. Thank you for pointing that out!

---

> > ### Comment · Reviewer_Qkax · 2023-08-18
> > **Response to the Rebuttal**
> >
> > Thank you for your response. Most of my questions have been resolved after reading the rebuttal.
> >
> > I will increase my score to 5.

---

> > > ### Author Response · Authors · 2023-08-18
> > > **thank you**
> > >
> > > We are glad to hear that our response helped resolved your questions - thank you again for your time and constructive feedback!

---

> ### Author Response · Authors · 2023-08-17
> **Thanks for your feedback**
>
> Dear Reviewer Qkax,
>
> As the discussion period ends soon, we just wanted to check if the response clarified your questions. Thanks again for your constructive feedback.
>
> Best,
>
> Authors

---

### Author Rebuttal · Authors · 2023-08-08

We thank all the reviewers for their time and valuable comments. We appreciate that reviewers find our approach DREAM-OOD **novel** and **effective** (Qkax, 94Wq, 3rTL), and the results are **comprehensive**, **extensive** and **promising** with **detailed** and **sufficient** ablations (Qkax, 94Wq, thbj, 3rTL, i54j). We are glad that all reviewers found the paper **well-written** and **easy to follow** (Qkax, 94Wq, thbj, 3rTL, i54j).

As endorsed by multiple reviewers, the work makes important contributions to the field for the following reasons:

- DREAM-OOD is the first to generate photo-realistic high-resolution outlier images for OOD detection by way of diffusion models. DREAM-OOD advances the field by helping researchers visualize and understand the imagined outliers in the pixel space, which was impossible in the past with feature-based synthesis methods (e.g., VOS or NPOS).
- Moreover, DREAM-OOD offers strong empirical performance with extensive results, which are recognized by all reviewers.
- Lastly, going beyond outlier imagination, our framework can be extended to generate inliers to improve model generalization. We demonstrate stronger performance than some of the most popular data augmentation methods, which we believe will benefit an even broader community and ML applications.

We respond to each reviewer's comments in detail below. We will also revise the manuscript according to the reviewers' suggestions, and we believe this makes our paper much stronger.

---

### Decision · Program_Chairs · 2023-09-21

**Decision:**

Accept (poster)

**Comment:**

This paper studies the setting of generating out-of-distribution data with diffusion models. The model is trained with in-distribution data and then low-density latent samples are obtained to generate OOD data. The reviews are unanimous, and the rebuttal have addressed the concerns of the reviewers. Acceptance is recommended.